# Cross-asset momentum and the hybrid fund transmission mechanism in China's stock and bond markets

Xiaowei Wang[1]*, Rui Wang[1], Yichun Zhang[2]

1 School of Economics, Xihua University, Chengdu, Sichuan, China, 2 School of Economics, Xiamen University, Xiamen, Fujian, China

* wangzp510719@163.com

## Abstract

The allocation of assets across different markets is a crucial element of investment strategy. In this regard, stocks and bonds are two significant assets that form the backbone of multi-asset allocation. Among publicly offered funds (The publicly offered funds in China correspond to the mutual funds in the United States, with different names and details in terms of legal form and sales channels), the stock-bond hybrid fund gives investors a return while minimizing the risk through capital flow between the stock and bond markets. Our research on China's financial market data from 2006 to 2022 reveals a cross-asset momentum between the stock and bond markets. We find that the momentum in the stock market negatively influences the bond market's return, while the momentum in the bond market positively influences the stock market's return. Portfolios that exploit cross-asset momentum have excess returns that other asset pricing factors cannot explain. Our analysis reveals that hybrid funds play an intermediary role in the transmission mechanism of cross-asset momentum. We observe that the more flexible the asset allocation ratio of the fund, the more crucial the intermediary role played by the fund. Hence, encouraging the development of hybrid funds and relaxing restrictions on asset allocation ratios could improve liquidity and pricing efficiency. These findings have significant implications for investors seeking to optimize their asset allocation across different markets and for policymakers seeking to enhance the efficiency of China's financial market.

## 1. Introduction

China's financial market has witnessed remarkable growth in recent years. As of July 2022, the total market capitalization of China's A-share market had surged to approximately 12 trillion US dollars, solidifying its position as one of the world's largest stock markets. Furthermore, China's bond market has expanded significantly, boasting a staggering size of 20 trillion US dollars, thereby establishing itself as the second-largest bond market worldwide (The afore-mentioned market data is sourced from the World Bank and the WIND database. Wind Data-base is a large-scale data warehouse with China's financial securities data as the core, covering

**Funding:** The authors received no specific funding for this work.

**Competing interests:** NO authors have competing interests.

stocks, funds, bonds, foreign exchange, insurance, futures, financial derivatives, spot transactions, macroeconomics. financial news and other fields). These developments have imparted substantial influence on the global financial markets. With the continuous advancement of interest rate liberalization in China, the link between the stock and bond markets has strengthened, leading to a favorable interaction between the two markets and facilitating optimal capital allocation.

As institutional investors, publicly offered funds play a crucial role in China's financial market. The total number of publicly offered funds issued in China has surpassed 10,000, with an overall size exceeding 3.7 trillion US dollars as of the first half of 2022, as illustrated in Figs 1 and 2. The explosive growth of stock-bond hybrid funds is particularly noticeable, reflecting changes in investor preferences toward diversified asset allocation. In contrast to single asset-focused funds, hybrid funds balance the return and the risk through investments in the stock and bond markets. Despite the stock crash event in 2015 (China's stock crash in 2015 was an event in which the stock market indexes plummeted in a short period of time from June to July 2015), the number of hybrid funds has not decreased but increased, exceeding the total number of both stock and bond funds. Moreover, hybrid funds hold the largest total size in China. The growth of hybrid funds in number and size demonstrates the evolution of China's investor preferences toward a more balanced and diversified asset allocation strategy.

Investors are faced with the critical issue of selecting an asset allocation strategy. Brinson et al. (1991) [1] and Ibbotson & Kaplan (2000) [2] conducted a study of over 200 mutual funds and found that the proportion of asset allocation explained 90% of the variance of the mutual fund's return. Based on these conclusions, tactical and strategic asset allocation concepts have been derived. Tactical asset allocation involves investing in specific asset types and relies on selecting different individuals to go long or short under the same type of assets, such as stock

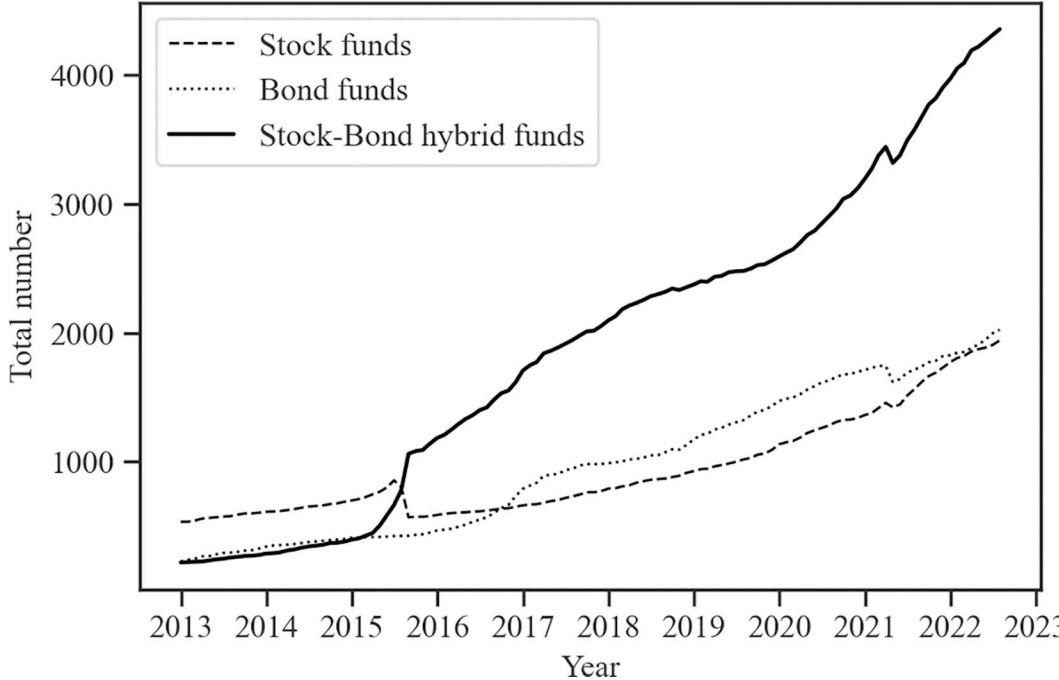

**Fig 1. The total number of stock, bond, and stock-bond hybrid funds in China.** Fig 1 displays the time series of the total number of different types of funds. The data comes from the Wind Database. The sample period is January 2013 through December 2022.

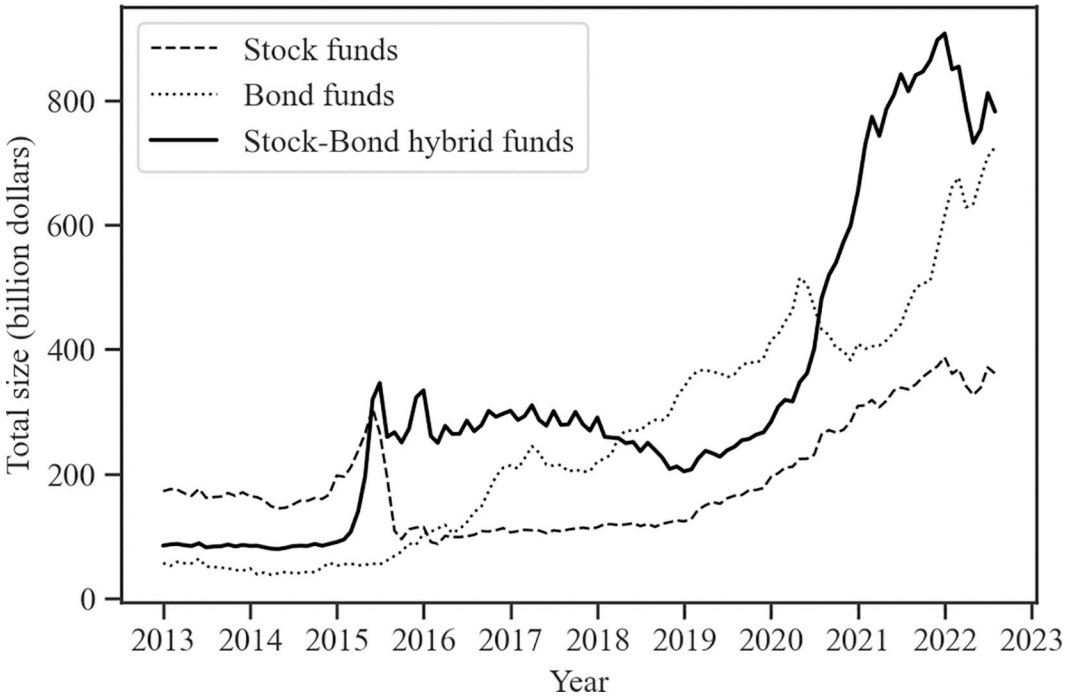

**Fig 2. The total size of stock, bond, and stock-bond hybrid funds in China.** Fig 2 displays the time series of the total size (in billion dollars) of different types of funds. The data comes from the Wind Database. The sample period is January 2013 through December 2022.

and bond funds limited by the asset allocation ratio. On the other hand, strategic asset allocation emphasizes adjusting asset allocation ratios based on changes in market fundamentals, such as hybrid funds with fewer restrictions on asset allocation ratios.

As an emerging market, it is essential to investigate the linkage mechanism between China's stock and bond markets and determine how investors can use it to build investment portfolios. Additionally, we are interested in the role of publicly offered funds (especially hybrid funds) as crucial institutional investors in the linkage.

Our study analyzes the market data in China from 2006 to 2022 to investigate these issues. First, stock funds require fund managers with a solid ability to select stocks. However, in China, the average tenure of fund managers is less than three years (According to Wind Database, as of September 2022, China's mutual fund data shows that about 31% of fund managers have an average management experience of over 1 year, and about 64% have an average management experience of 1–5 years. This means that 95% of fund managers in the entire market have an average management experience of less than 5 years). This situation presents a challenge for investors seeking consistent and long-term returns. Additionally, individual investors tend to invest in bonds through publicly offered funds because direct bond investments often have high entry barriers. Unfortunately, bond funds usually offer low-risk, low-return characteristics that do not align with the high-return expectations of individual investors.

Furthermore, compared to the well-established financial markets of developed countries, there is a need to enhance the connection between China's stock and bond markets, as noted by Chen and Zeng in 2016 [3]. The investment behavior of hybrid funds in both markets has the potential to facilitate the flow of capital and information.

Moreover, it is vital to recognize the broader significance of studying the linkages within China's stock and bond markets. China's markets can serve as representative examples of

emerging markets. Emerging markets differ from their developed counterparts regarding investor composition (notably, a lower percentage of institutional investors) and investment culture (which significantly influence investor sentiments). Consequently, conclusions drawn from studies based on data from developed markets may not directly apply to emerging markets. Our research aims to fill this gap in understanding cross-market linkages within emerging markets.

Lastly, China's regulatory authorities impose strict regulations on institutional investors, specifying the types and proportions of their pre-investment allocations. As our subsequent findings in this paper demonstrate, different categories of institutional investors play distinct roles in cross-market mechanisms. This perspective is often overlooked by researchers primarily focusing on developed country markets.

## 2. Literature review and theoretical analysis

The linkage mechanism between China's stock and bond markets has been extensively studied in the literature. Previous studies have focused on the risk spillover effects between the two markets, with most studies concluding that the relationship between stocks and bonds is mainly risk hedging. Chen and Zeng (2016) [3] found that China's corporate bonds and stocks consistently exhibit significant tail risk spillover effects, while China's government bonds and stocks only display this relationship during bear market phases. Hou et al. (2020) [4] studied the risk spillover between China's stock and bond markets and found an asymmetry in the risk spillover. Zhou et al. (2020) [5] studied the hedging effect between China's stock and bond markets and showed that the cross-asset pricing mechanism conforms to skewness preference and kurtosis aversion assumptions.

The literature discussed above predominantly examines intermarket linkages in China's stock and bond markets, primarily from a risk-centric perspective. It is essential to acknowledge that the objectives of cross-market investors transcend risk mitigation. Many investors, including high-capitalization mutual funds, are deeply vested in pursuing returns through cross-market investments.

Whether categorized as stock, bond, or hybrid funds, a substantial segment of their asset allocations is directed toward the stock and bond markets. From the vantage point of strategic asset allocation, how funds apportion their capital across these markets significantly influences their ultimate performance.

Now, let us approach this issue from the standpoint of these investment vehicles. The net asset value of a fund, or, in more precise terms, the wealth of its investors, is inherently contingent on the valuations of stocks and bonds. After this valuation process, investors engage in decision-making. This paper introduces two fundamental hypotheses:

1. When investors possess more significant levels of wealth, their heightened risk tolerance inclines them toward an increased allocation to equities and a diminished allocation to bonds. This proclivity is rooted in the perception that equities, characterized by their high-risk, high-return nature, assume the attributes of a "luxury" during such periods. Investors, endowed with the capability to assume greater risk, are willing to pursue enhanced returns.

2. Conversely, when investors possess lower levels of wealth, their risk tolerance diminishes, leading them to favor a higher allocation to bonds and a reduced allocation to equities. In this scenario, low-risk, low-return bonds assume the role of a "necessity," and investors adopt a more conservative stance, gravitating towards bonds to preserve their capital.

Indeed, counterarguments posit that wealthier individuals may adopt a more conservative approach to safeguard their existing wealth, while those with lesser wealth may adopt more

aggressive investment strategies. Nevertheless, from an economic perspective, the earlier notion we have presented regarding "luxury" and "necessity" appears more conceptually sound. This perspective harmonizes with the intended direction of our forthcoming empirical results section in the paper.

Should our hypotheses hold, it can be anticipated that extended periods of consistent directional movements in stock and bond prices (short-term price fluctuations may not elicit immediate responses from institutional investors due to their extended decision-making horizons and associated costs) will exert an impact on investor wealth levels. As a consequence of these wealth fluctuations, funds will initiate capital reallocations between the stock and bond markets. Given the substantial influence of institutional investors within the market, we can reasonably envisage further shifts in asset prices. This conceptual framework evokes the notion of "momentum."

The asset momentum effect, which refers to price changes maintaining their direction, has been the focus of research in academic and investment fields since Jegadesh and Titman (1993) [6] first studied the stock momentum effect. As a long-term market anomaly, the momentum effect has attracted significant attention, and its causes have different interpretations. One explanation is the asset pricing factor framework, which attributes the return of momentum effect to the compensation obtained by investors for taking specific risks. Daniel and Moskowitz (2016) [7] contend that the tail risk in the distribution of returns is the source of compensation for momentum investors, which means that investors bear the risk that stocks that continue to rise in price have a probability of experiencing sharp price drops. Geczy and Samonov (2016) [8], on the other hand, view momentum investors as bearing the risk of a turning point in the economic cycle and, as such, receive compensation.

Another explanation of the momentum effect is from the perspective of behavioral finance, which argues that it comes from the incomplete rational behavior of investors. Daniel and David (1998) [9] believe that investors have limited cognitive abilities, leading to an underreaction or overreaction to market information, thus causing asset prices not to adjust immediately after the arrival of new information. Grinblatt and Han (2005) [10] explain momentum based on the psychological disposition effect, where investors cannot make immediate decisions even when they receive valid information due to loss aversion and conservative circumstances.

Moreover, the momentum effect has attracted attention because the momentum portfolio can obtain excess returns that other asset pricing factors cannot explain. Since Ross (1976) [11] proposed the arbitrage pricing theory, investors have studied various pricing factors to explain investment returns. While many factors have been proposed, few can pass the test of time and market (Feng et al. 2020) [12]. The momentum effect, however, is considered a long-term influential pricing factor (Asness et al. 2013) [13]. Moskowitz et al. (2012) [14] distinguished momentum into cross-sectional and time-series momentum. Cross-sectional momentum refers to the phenomenon that the returns of historical winner assets continue to exceed those of historical loser assets. Time-series momentum refers to the persistence of an asset's return in the time series. Moskowitz [14] believes that time-series momentum can explain cross-sectional momentum, not vice versa. Time-series momentum challenges Fama's (1970) [15] efficient market hypothesis, which holds that investors cannot profit from historical information. Time-series momentum, however, shows that historical returns can predict future returns.

As an emerging economy, China's financial market displays unique characteristics that differ from developed countries. Notably, the market is characterized by many individual investors (As of the end of October 2022, the number of investors in the China's A-share market reached 210 million, with individual investors accounting for 99.8% and individual investors accounting for 61.35% of the trading volume), relatively high transaction costs, and short-

selling restrictions (For example, in China, companies that can become the target stocks for securities lending often have stable operations and excellent performance. Stocks that are often easily identified as short-selling targets in developed markets are not allowed to engage in margin trading in China). Consequently, the performance of momentum effects in the market differs from those observed in developed countries. Specifically, studies such as Lu and Zou (2007) [16], Pan and Xu (2011) [17], and Gao et al. (2014) [18] have shown that the momentum effect in China's stock market is challenging to sustain in the medium and long term and is characterized by a short-term reversal. The weak momentum effect in China's stock market is attributed to several factors. Chen et al. (2014) [19] and Zhu et al. (2017) [20] contend that the restrictions on short selling and the high transaction costs deduct the momentum returns. Furthermore, Bai et al. (2020) [21] reveal that the "T+1" system significantly affects the momentum effect in the market (China's stock market implements a T+1 trading system, where stocks purchased on the same day cannot be sold until the next trading day). Their study finds that the intraday and overnight momentum returns under the constraint system have opposite effects that offset the overall momentum returns.

Several studies have investigated the performance of momentum returns in China's stock market, and most of them suggest that it is difficult for investors to obtain such returns. However, some studies have also highlighted the momentum trading behavior of institutional investors in China's stock market. Chen et al. (2008) [22] found that institutional investors tend to adopt positive feedback trading strategies, while individual investors exhibit more random trading behavior. To reconcile these seemingly contrasting findings, Wang et al. (2020) [23] proposed a "new momentum effect" based on the sample of China's stock funds. The study showed that after excluding the top 1%-5% of funds with the best recent performance, the remaining funds' portfolios exhibit a more significant and stable momentum effect. Furthermore, Tang (2022) [24] investigated the momentum trading behavior of actively managed stock funds and found significant momentum trading within the past year's performance as the lookback period.

It is important to note that many individual investors in China's stock market may influence the performance of the momentum effect. These noise traders' short-term trading behavior may affect the momentum effect's sustainability. In contrast, institutional investors can adhere to momentum investment strategies in the medium and long term but may not have a significant influence on the overall market.

In conclusion, while the weak momentum effect in China's A-share market is a widely accepted phenomenon, studies have also highlighted the momentum trading behavior of institutional investors. The presence of noise traders in the market and institutional investors' cross-market asset allocation may partially explain the coexistence of these seemingly contrasting findings. Further research is needed to explore these findings' underlying mechanisms and implications.

While several studies have explored the momentum effect in China's stock market and the momentum trading of stock funds, limited research has investigated the linkage between the momentum effect of China's stock market and bond markets and the momentum trading behavior of hybrid funds. In reality, the stock and bond markets are highly interrelated, with a persistent flow of information and capital between the two markets. Thus, continuous changes in one asset's price will influence another's price.

Research-based on developed financial markets provides valuable insights into explaining the factors behind cross-asset momentum effects. Duffie (2010) [25] delves into the theory of financial market frictions and slow capital flows, uncovering that changes in the future demand for stocks and bonds induced by their historical returns over the past 12 months occur slowly over the next few months. One of the contributing factors to this phenomenon is

the presence of capital frictions, such as the time required for investment decisions and the gradual resolution of massive capital outflows. Greenwood et al. (2018) [26] explore how demand and supply shocks in one market affect another in relatively segmented asset markets. They find that asset prices tend to exhibit overreactions in markets directly impacted by shocks. In contrast, markets not directly affected by these shocks demonstrate relatively delayed responses due to the actions of arbitrageurs reallocating assets within the market. Ben-Rephael et al.(2011) [27] (2012) [28] investigated the relationship between stock fund trading volume and stock market indices, revealing that fluctuations in fund flows can predict some of the future returns in the stock market. This conclusion aligns with economic intuition, particularly in today's environment with the growing dominance of institutional investors. We are particularly interested in understanding whether the research findings above apply to emerging markets like China, especially with the recent emergence of hybrid funds, and how they contribute to our comprehension of cross-asset momentum.

In addition to stock funds, other types of funds, especially hybrid funds that have emerged with the concept of multi-asset allocation in recent years, play a significant role in China's financial market. These hybrid funds hold a considerable share of investments and are important market participants. Therefore, it is necessary to investigate the cross-asset momentum between China's stock and bond markets and analyze the role of different types of funds in momentum transmission, which can provide a more comprehensive understanding of the market's dynamics and enhance investment decision-making.

Our research has unearthed critical insights and made two significant contributions to the literature. First, we have identified a noteworthy cross-asset momentum between the stock and bond markets. Notably, we find that momentum in the stock market exerts a negative influence on bond market returns, while momentum in the bond market positively impacts stock market returns. Additionally, portfolios designed to exploit this cross-asset momentum exhibit excess returns that cannot be accounted for by conventional asset pricing factors. Second, Our analysis also highlights the pivotal role of hybrid funds in transmitting cross-asset momentum. Significantly, we observe that the flexibility of asset allocation ratios within these funds corresponds to the extent of their intermediary role.

## 3. Methodology

We investigate the existence of cross-asset momentum between China's stock and bond markets and the role of different types of funds in momentum transmission. The research methodology consists of three steps.

Firstly, we examine whether a significant cross-asset momentum exists. We use the weekly frequency data of the CSI 300 Index (The CSI 300 Index is a constituent stock index compiled and published by China Securities Index Corporation, which selects 300 A-share's stocks with high market value and good liquidity in the Shanghai and Shenzhen securities markets, weighted by the market value of freely tradable equity) and the SSE Corporate Bond Index (The Shanghai Stock Exchange Corporate Bond Index is composed of representative corporate bonds that meet certain conditions and are listed on the Shanghai and Shenzhen Stock Exchanges, reflecting the overall performance of the Shanghai and Shenzhen corporate bond market) to represent the stock and bond market indexes, respectively. The CSI 300 Index comprises 300 representative stocks with enormous market value and high liquidity in China. The SSE Corporate Bond Index comprises representative bonds among the corporate bonds listed on the Shanghai Stock Exchange.

Secondly, we construct a stock and bond market index portfolio based on the cross-asset momentum strategy and examine whether the portfolio generates a significant excess return.

**Table 1. Classification criteria of fund indexes.**

| Index | Type | Stock ratio | Bond ratio |
|---|---|---|---|
| **CSI Fund Index** | Stock type | > 80% net assets | - |
| | Bond type | - | > 80% net assets |
| | Mixed type | < 80% net assets | < 80% net assets |
| **Wind Fund Index** | Stock type | > 90% net assets | - |
| | Partial Stock type | > 60% net assets | - |
| | Bond type | - | > 90% net assets |
| | Partial Bond type | - | > 60% net assets |
| | Flexible type | < 90% net assets | < 90% net assets |

This table reports the classification criteria of the CSI Fund Index and the Wind Fund Index. According to the allocation ratio of assets, the CSI Fund Index is classified into three types: Stock type, Bond type, and Mixed stock-bond type, and the Wind Fund Index is classified into five types: Stock type, Partial Stock type, Bond type, Partial Bond type, and Flexible type.

Finally, we explore the role of different types of funds in cross-asset momentum transmission, with the CSI Fund Index and the Wind Fund Index representing different types of funds. These fund indexes are composed of the weighted method of the net settlement value on the current trading day, reflecting the overall value changes of different types of funds in the market. The CSI Fund Index is classified into three categories: Stock type, Bond type, and Mixed type based on the asset allocation ratio, while the Wind Fund Index is classified into five categories: Stock type, Partial stock type, Bond type, Partial Bond type, and Flexible type. The specific classification criteria are presented in Table 1.

### 3.1. Cross-asset momentum with OLS and quantile regression

**3.1.1. OLS regression analysis.** We employ the OLS regression model to investigate the cross-asset momentum between China's stock and bond markets. The model used in this study is adapted from Moskowitz et al. (2012) [14] and comprises two equations as follows:

$$\frac{STOCK_{t,t+4}}{\sigma_{t-52,t-1}^{STOCK}} = \beta_0 + \beta_1 \frac{BOND_{t-k,t+4-k}}{\sigma_{t-52-k,t-1-k}^{BOND}} + \beta_2 \frac{STOCK_{t-k,t+4-k}}{\sigma_{t-52-k,t-1-k}^{STOCK}} + \varepsilon_t \tag{1}$$

$$\frac{BOND_{t,t+4}}{\sigma_{t-52,t-1}^{BOND}} = \beta_0 + \beta_1 \frac{STOCK_{t-k,t+4-k}}{\sigma_{t-52-k,t-1-k}^{STOCK}} + \beta_2 \frac{BOND_{t-k,t+4-k}}{\sigma_{t-52-k,t-1-k}^{BOND}} + \varepsilon_t \tag{2}$$

Eq (1) estimates the cross-asset momentum effect of the bond market on the stock market, while Eq (2) estimates the cross-asset momentum effect of the stock market on the bond market.

$STOCK_{t,t+4}$ and $BOND_{t,t+4}$ represent the risk-returns of the CSI 300 Index and the SSE Corporate Bond Index in the subsequent four trading weeks, equivalent to one trading month. Risk returns are calculated by subtracting the risk-free rate from the nominal return of the market index, and we select the 3-month fixed deposit benchmark interest rate of the Central Bank of China as the risk-free rate.

Similarly, $STOCK_{t-k,t+4-k}$ and $BOND_{t-k,t+4-k}$ denote the risk-returns of the CSI 300 Index and the SSE Corporate Bond Index in the past $k$ trading weeks.

Furthermore, $\sigma_{t-52,t-1}^{STOCK}$ and $\sigma_{t-52,t-1}^{BOND}$ represent the historical volatility of the risk-returns of the CSI 300 Index and the SSE Corporate Bond Index in the past 52 trading weeks.

$\beta_1$ represents the influence of the historical cumulative return of one asset on the future return of another asset, that is, the cross-asset momentum. We estimate $\beta_1$ for the past 1 to $k$ trading weeks. If $\beta_1$ has the same sign, and certain coefficients are statistically significant during a specific period, we consider that there is cross-asset momentum between the two markets.

$\beta_2$ represents the influence of the asset's historical cumulative return on its future return and controls its time-series momentum. The intercept term is denoted by $\beta_0$, and $\varepsilon_t$ represents the residual term.

Following the model framework established by Moskowitz et al. (2012), we adjust returns by volatility for the following reasons: from a statistical perspective, this adjustment approximates a form of generalized least squares, which mitigates the autocorrelation of return series residuals. In economic terms, it approximates using the Sharpe ratio instead of raw returns, thereby enhancing the efficiency of estimating asset performance.

**3.1.2. Quantile regression analysis.** We also employ the quantile regression model to investigate the cross-asset momentum:

$$STOCK_{t,t+4} = \beta_0 + \beta_1 CUMBOND_{t-52,t-1} + \beta_2 CUMSTOCK_{t-52,t-1} + \beta_3 SHIBOR_{t,t+4}$$
$$+ \beta_4 M2_{t,t+4} + \beta_5 CPI_{t,t+4} + \beta_6 PMI_{t,t+4} + \varepsilon_t \tag{3}$$

$$BOND_{t,t+4} = \beta_0 + \beta_1 CUMSTOCK_{t-52,t-1} + \beta_2 CUMBOND_{t-52,t-1} + \beta_3 SHIBOR_{t,t+4} + \beta_4 M2_{t,t+4}$$
$$+ \beta_5 CPI_{t,t+4} + \beta_6 PMI_{t,t+4} + \varepsilon_t \tag{4}$$

Eq (3) estimates the cross-asset momentum effect of the bond market on the stock market, while Eq (4) estimates the cross-asset momentum effect of the stock market on the bond market. Considering the fat-tail distribution of returns, we perform regressions at different quantiles to enhance the robustness of the estimation.

$CUMSTOCK_{t-52,t-1}$ and $CUMBOND_{t-52,t-1}$ represent the historical cumulative returns of the CSI 300 Index and the SSE Corporate Bond Index in the past 52 trading weeks.

$\beta_1$ represents the influence of the historical cumulative return of one asset on the future return of another asset, that is, the cross-asset momentum effect.

$\beta_2$ represents the influence of the asset's historical cumulative return on its future return and serves as the control variable for time series momentum. $\beta_0$ represents the intercept term and $\varepsilon_t$ represents the residual term.

To control for the influence of macro policies and market fundamentals on asset prices, we select $SHIBOR_{t,t,+4}$ to represent the Shanghai Interbank Offered Rate, $M2_{t,t,+4}$ represents the M2 money supply, $CPI_{t,t,+4}$ represents the change rate of the consumer price index and $PMI_{t,t,+4}$ represents the rate of change of the Purchasing Managers' Index. We select these control variables from Zheng and Xu's (2020) [29] research.

## 3.2. Momentum portfolio analysis

We construct single-asset and cross-asset momentum portfolios with the CSI 300 Index and the SSE Corporate Bond Index and employ asset pricing factors to examine the excess return.

**3.2.1. Single-asset momentum portfolio analysis.** The return expressions of the single-asset momentum portfolio are as follows:

When short selling is prohibited:

$$TSMOM_{t,t+4} = [I(CUMSTOCK_{t-52,t-1} > 0)STOCK_{t,t+4} + I(CUMBOND_{t-52,t-1}$$
$$> 0)BOND_{t,t+4}]/2 \qquad (5)$$

When short selling is allowed:

$$TSMOM_{t,t+4} = [sign(CUMSTOCK_{t-52,t-1} > 0)STOCK_{t,t+4} + sign(CUMBOND_{t-52,t-1}$$
$$> 0)BOND_{t,t+4}]/2 \qquad (6)$$

$TSMOM_{t,t+4}$ represents the single-asset momentum portfolio's one-month risk return during the holding period.

Eq (5) represents the portfolio's risk-return when short selling is prohibited. $I(*)$ is an indicator function and $I(CUMSTOCK_{t-52,t-1} > 0)STOCK_{t,t+4}$ represents the risk-return of the stock position. Specifically, when the historical cumulative risk-return of the CSI 300 Index in the past 52 trading weeks is greater than 0, we long and hold the CSI 300 Index for the next four weeks. In contrast, when the cumulative risk-return is less than 0, we invest in risk-free assets (e.g., cash) with a zero risk return. $I(CUMBOND_{t-52,t-1} > 0)BOND_{t,t+4}$ represents the risk return of the bond position. The initial investment position is equally divided between the stock and bond positions and rebalanced at the end of each period.

Eq (6) represents the long-short return of the portfolio under the condition of allowing short selling. Here, $sign(*)$ is a sign function, and $sign(CUMSTOCK_{t-52,t-1} > 0)STOCK_{t,t+4}$ represents the risk-return of the stock position. When the cumulative risk-return of the CSI 300 Index in the past 52 weeks is greater than 0, we long and hold the CSI 300 Index for the next four weeks. In contrast, we short the CSI 300 Index when the cumulative risk-return is less than or equal to 0. The initial investment position is equally divided between the stock and bond positions and rebalanced at the end of each period.

**3.2.2. Cross-asset momentum portfolio analysis.** The return expressions of the cross-asset momentum portfolio are as follows:

When short selling is prohibited:

$$XTSMOM_{t,t+4} = [I(CUMSTOCK_{t-52,t-1} > 0, CUMBOND_{t-52,t-1}$$
$$> 0)STOCK_{t,t+4} + I(CUMBOND_{t-52,t-1} > 0, CUMSTOCK_{t-52,t-1}$$
$$< 0)BOND_{t,t+4}]/2 \qquad (7)$$

When short selling is allowed:

$$XTSMOM_{t,t+4} = [sign(CUMSTOCK_{t-52,t-1} > 0, , CUMBOND_{t-52,t-1}$$
$$> 0)STOCK_{t,t+4} + sign(CUMBOND_{t-52,t-1} > 0, CUMSTOCK_{t-52,t-1}$$
$$< 0)BOND_{t,t+4}]/2 \qquad (8)$$

$XTSMOM_{t,t+4}$ represents the cross-asset momentum portfolio's one-month risk return during the holding period.

Eq (7) represents the portfolio's return when short selling is prohibited, and Eq (8) represents the long-short return of the portfolio when short selling is allowed. Compared to single-asset momentum strategies, the cross-asset momentum strategy considers not only the information of the historical cumulative return of an individual asset but also the information of the historical cumulative return of another asset.

For example, $I(CUMSTOCK_{t-52,t-1} > 0, CUMBOND_{t-52,t-1} > 0)STOCK_{t,t+4}$ indicates that the investor will only long the CSI 300 Index if the historical cumulative risk-return of the CSI 300 Index in the past 52 trading weeks is greater than 0 and the cumulative historical risk-return of the SSE Corporate Bond Index in the past 52 trading weeks is also greater than 0. Otherwise, the investor will hold risk-free assets.

**3.2.3. Risk-adjusted performance of the momentum portfolio.** We use the asset pricing factor model to examine whether there exists an excess return in the momentum portfolio, and the regression models are as follows:

$$r_{t,t+4} = Alpha_{t,t+4} + \beta_1 MKT_{t,t+4} + \beta_2 SMB_{t,t+4} + \beta_3 HML_{t,t+4} + \beta_4 RMW_{t,t+4} + \beta_5 CMA_{t,t+4}$$
$$+ \beta_6 UMD_{t,t+4} + \varepsilon_t \tag{9}$$

$$r_{t,t+4} = Alpha_{t,t+4} + \beta_1 Level_{t,t+4} + \beta_2 Slope_{t,t+4} + \beta_3 Curvature_{t,t+4} + \varepsilon_t \tag{10}$$

$r_{t,t,+4}$ represents the risk-return of the portfolio during the holding period and $Alpha_{t,t,+4}$ represents the excess return of the portfolio after controlling other pricing factors.

Eq (9) is the stock pricing factor model, referring to Li's (2017) [30] research on the pricing factors of China's A-share market, $MKT_{t,t,+4}$ represents the market risk factor, $SMB_{t,t,+4}$ represents the size factor, $HML_{t,t,+4}$ represents the value factor, $RMW_{t,t,+4}$ represents the quality factor, and $CMA_{t,t,+4}$ represents the investment style factor, $UMD_{t,t,+4}$ represents the cross-sectional momentum factor (The stock pricing factors come from the CSMAR database, see https://cn.gtadata.com / for more details).

Eq (10) is the bond pricing factor model, which relies on Tang and Zhu (2003) [31] and Shang and Zheng's (2018) [32] three-factor model of the yield curve to analyze the return of China's bond market. $Level_{t,t,+4}$ is the level factor, representing the corporate bond yield curve's long-term (e.g., ten-year) interest rate. $Slope_{t,t,+4}$ is the slope factor, which represents the interest rate difference between the long-term and short-term (e.g., one year) of the corporate bond yield curve. $Curvature_{t,t,+4}$ is the curvature factor, which represents the second derivative of the corporate bond yield curve, that is, the curvature. $\varepsilon_t$ represents the residual term.

## 3.3. Cross-asset momentum transmission mechanism

**3.3.1. Momentum from the bond to the stock market.** We employ the following regression models to investigate the intermediary role of hybrid funds in transmitting cross-asset momentum from the bond market to the stock market. Eqs (11) and (12) are OLS regressions, while Eq (13) is a two-stage instrumental variable (IV-2SLS) regression.

$$FUND_{t,t+4} = \alpha_0 + \alpha_1 CUMBOND_{t-52,t-1} + \alpha_2 CUMSTOCK_{t-52,t-1} + \alpha_3 SHIBOR_{t,t+4} + \alpha_4 M2_{t,t+4}$$
$$+ \alpha_5 CPI_{t,t+4} + \alpha_6 PMI_{t,t+4} + \varepsilon_t \tag{11}$$

$$STOCK_{t,t+4} = \beta_0 + \beta_1 CUMBOND_{t-52,t-1} + \beta_2 CUMSTOCK_{t-52,t-1} + \beta_3 SHIBOR_{t,t+4}$$
$$+ \beta_4 M2_{t,t+4} + \beta_5 CPI_{t,t+4} + \beta_6 PMI_{t,t+4} + \varepsilon_t \tag{12}$$

$$STOCK_{t,t+4} = \gamma_0 + \gamma_1 FUND_{t,t+4} + \gamma_2 CUMBOND_{t-52,t-1} + \gamma_3 SHIBOR_{t,t+4} + \gamma_4 M2_{t,t+4}$$
$$+ \gamma_5 CPI_{t,t+4} + \gamma_6 PMI_{t,t+4} + \varepsilon_t \tag{13}$$

Eq (11) estimates the influence of the bond and the stock indexes' historical cumulative return on the fund index's return and $FUND_{t,t+4}$ represents the risk return of the fund index.

Eq (12) estimates the influence of the bond and the stock indexes' historical cumulative return on the stock index's return.

If $\alpha_2$ in Eq (11) is significant, but $\beta_2$ in Eq (12) is not significant, the result indicates a direct correlation between $CUMSTOCK_{t-52,t-1}$ and $FUND_{t,t+4}$, but not $STOCK_{t,t+4}$. In other words, $CUMSTOCK_{t-52,t-1}$ may affect $STOCK_{t,t+4}$ through $FUND_{t,t+4}$. This is in line with our view on the momentum of China's stock market: the Time-series momentum of the stock market is generally weak, but institutional investors such as hybrid funds may adopt momentum investment strategies.

Eq (13) estimates the influence of the fund index's return on the stock index's return. To address the endogeneity between $FUND_{t,t+4}$ and $STOCK_{t,t+4}$, we select $CUMSTOCK_{t-52,t-1}$ as the instrumental variable.

Finally, if $\alpha_1$ in Eq (11), $\beta_1$ in Eq (12), $\gamma_1$ and $\gamma_2$ in Eq (13) are significant, it implies that the fund may have played an intermediary role in transmitting the bond market's momentum to the stock market. $\alpha_0$, $\beta_0$ and $\gamma_0$ represent the intercept and $\varepsilon_t$ represents the residual item. $SHIBOR_{t,t+4}$, $M2_{t,t+4}$, $CPI_{t,t+4}$ and $PMI_{t,t+4}$ are control variables.

**3.3.2. Momentum from the stock to the bond market.** We employ the following regression models to investigate the intermediary role of funds in transmitting cross-asset momentum from the stock market to the bond market. Eqs (14) and (15) are OLS regressions, while Eq (16) is a two-stage instrumental variable (IV-2SLS) regression.

$$FUND_{t,t+4} = \alpha_0 + \alpha_1 CUMSTOCK_{t-52,t-1} + \alpha_2 CUMBOND_{t-52,t-1} + \alpha_3 SHIBOR_{t,t+4} + \alpha_4 M2_{t,t+4}$$
$$+ \alpha_5 CPI_{t,t+4} + \alpha_6 PMI_{t,t+4} + \varepsilon_t \tag{14}$$

$$BOND_{t,t+4} = \beta_0 + \beta_1 CUMSTOCK_{t-52,t-1} + \beta_2 CUMBOND_{t-52,t-1} + \beta_3 SHIBOR_{t,t+4} + \beta_4 M2_{t,t+4}$$
$$+ \beta_5 CPI_{t,t+4} + \beta_6 PMI_{t,t+4} + \varepsilon_t \tag{15}$$

$$BOND_{t,t+4} = \gamma_0 + \gamma_1 FUND_{t,t+4} + \gamma_2 CUMSTOCK_{t-52,t-1} + \gamma_3 SHIBOR_{t,t+4} + \gamma_4 M2_{t,t+4}$$
$$+ \gamma_5 CPI_{t,t+4} + \gamma_6 PMI_{t,t+4} + \varepsilon_t \tag{16}$$

Eq (14) estimates the influence of the stock and bond indexes' historical cumulative returns on the fund index's returns.

Eq (15) estimates the influence of the stock and the bond indexes' historical cumulative return on the bond index's return.

If $\alpha_2$ in Eq (14) is significant, but $\beta_2$ in Eq (15) is not significant, the result indicates a direct correlation between $CUMBOND_{t-52,t-1}$ and $FUND_{t,t+4}$, but not $BOND_{t,t+4}$. $CUMBOND_{t-52,t-1}$ may affect $BOND_{t,t+4}$ through $FUND_{t,t+4}$.

Eq (16) estimates the influence of the fund index's return on the bond index's return. To address the endogeneity between $FUND_{t,t+4}$ and $BOND_{t,t+4}$, we select $CUMBOND_{t-52,t-1}$ as the instrumental variable.

Finally, if $\alpha_1$ in Eq (14), $\beta_1$ in Eq (15), $\gamma_1$ and $\gamma_2$ in Eq (16) are significant, it implies that the fund may have played an intermediary role in transmitting the stock market's momentum to the bond market.

## 4. Empirical results

### 4.1. Data and descriptive statistics

The descriptive statistics of the main variables are shown in Table 2. We utilize weekly frequency data of the CSI 300 Index, the SSE Corporate Bond Index, and different types of Fund

**Table 2. Summary statistics of main variables.**

| Variable | N | Mean | Std.Dev. | Min | Max |
|---|---|---|---|---|---|
| $STOCK_{t,t+4}$ | 813 | 0.008 | 0.090 | -0.310 | 0.297 |
| $BOND_{t,t+4}$ | 813 | 0.003 | 0.008 | -0.029 | 0.083 |
| $FUND_{t,t+4}$ | | | | | |
| CSI Stock Fund Index | 813 | 0.011 | 0.086 | -0.373 | 0.520 |
| CSI Mixed Fund Index | 813 | 0.012 | 0.072 | -0.218 | 0.479 |
| CSI Bond Fund Index | 813 | 0.004 | 0.015 | -0.051 | 0.183 |
| Wind Stock Fund Index | 813 | 0.014 | 0.091 | -0.454 | 0.579 |
| Wind Partial Stock Fund Index | 813 | 0.012 | 0.084 | -0.368 | 0.533 |
| Wind Flexible Fund Index | 813 | 0.009 | 0.067 | -0.233 | 0.511 |
| Wind Partial Bond Fund Index | 813 | 0.007 | 0.030 | -0.086 | 0.277 |
| Wind Bond Fund Index | 813 | 0.004 | 0.014 | -0.043 | 0.161 |
| $MKT_{t,t+4}$ | 813 | 0.008 | 0.079 | -0.250 | 0.255 |
| $SMB_{t,t+4}$ | 813 | 0.008 | 0.049 | -0.222 | 0.227 |
| $HML_{t,t+4}$ | 813 | -0.001 | 0.033 | -0.140 | 0.157 |
| $RMW_{t,t+4}$ | 813 | -0.001 | 0.033 | -0.108 | 0.123 |
| $CMA_{t,t+4}$ | 813 | 0.002 | 0.023 | -0.066 | 0.081 |
| $UMD_{t,t+4}$ | 813 | 0.005 | 0.050 | -0.193 | 0.123 |
| $Level_{t,t+4}$ | 813 | 0.026 | 1.148 | -2.833 | 4.804 |
| $Slope_{t,t+4}$ | 813 | 0.072 | 1.084 | -2.569 | 4.700 |
| $Curvature_{t,t+4}$ | 813 | 0.022 | 0.793 | -3.865 | 2.688 |
| $SHIBOR_{t,t+4}$ | 813 | 0.003 | 0.001 | 0.001 | 0.006 |
| $M2_{t,t+4}$ | 813 | 0.014 | 0.023 | -0.012 | 0.308 |
| $CPI_{t,t+4}$ | 813 | 0.010 | 0.023 | -0.057 | 0.130 |
| $PMI_{t,t+4}$ | 813 | -0.012 | 0.046 | -0.340 | 0.376 |

This table reports summary statistics of the main variables. $STOCK_{t,t+4}$ is the monthly risk return of the CSI 300 Index. $BOND_{t,t+4}$ is the monthly risk return of the SSE Corporate Bond Index $FUND_{t,t+4}$ is the monthly risk return of different types of Fund Indexes. $MKT_{t,t+4}$, $SMB_{t,t+4}$, $HML_{t,t+4}$, $RMW_{t,t+4}$, $CMA_{t,t+4}$ and $UMD_{t,t+4}$ are the stock pricing factors from Li et al. (2017) [29]. $Level_{t,t+4}$, $Slope_{t,t+4}$ and $Curvature_{t,t+4}$ are the bond pricing factors from Shang and Zheng (2018) [32]. $SHIBOR_{t,t+4}$ is the one-month Shanghai Interbank Offered Rate $M2_{t,t+4}$ is the monthly growth rate of the M2 money supply. $CPI_{t,t+4}$ is the monthly growth rate of the Consumer Price Index. $PMI_{t,t+4}$ is the monthly growth rate of Purchasing Managers' Index. The sample period is October 2006 through August 2022.

Indexes from October 2006 to August 2022, encompassing 813 trading weeks. The stock and bond pricing factors are sourced from the CSMAR database, whereas the other control variables, including SHIBOR, M2, CPI, and PMI, are obtained from the Wind database.

## 4.2. Results of cross-asset momentum with OLS and quantile regression

**4.2.1. Results of cross-asset momentum with OLS regression.** In Figs 3 and 4, we report the result of Eq (1), where the horizontal axis corresponds to the past month, and the vertical axis corresponds to the $t$-statistics of the $\beta_1$ and the $\beta_2$ clustered by month.

Fig 3 reports the $t$-statistic of the $\beta_1$, and it is positive over the past 1 to 11 months, implying that the future return of the CSI 300 Index has a positive correlation with the historical returns of the SSE Corporate Bond Index.

Fig 4 reports the $t$-statistic of the $\beta_2$, which is positive over the past 1 to 5 months, indicating that the future return of the CSI 300 Index is also positively correlated with its historical returns.

Notably, the correlation duration between the future return of the CSI 300 Index and the historical returns of the SSE Corporate Bond Index is longer than the correlation duration between the future return of the CSI 300 Index and its own historical returns.

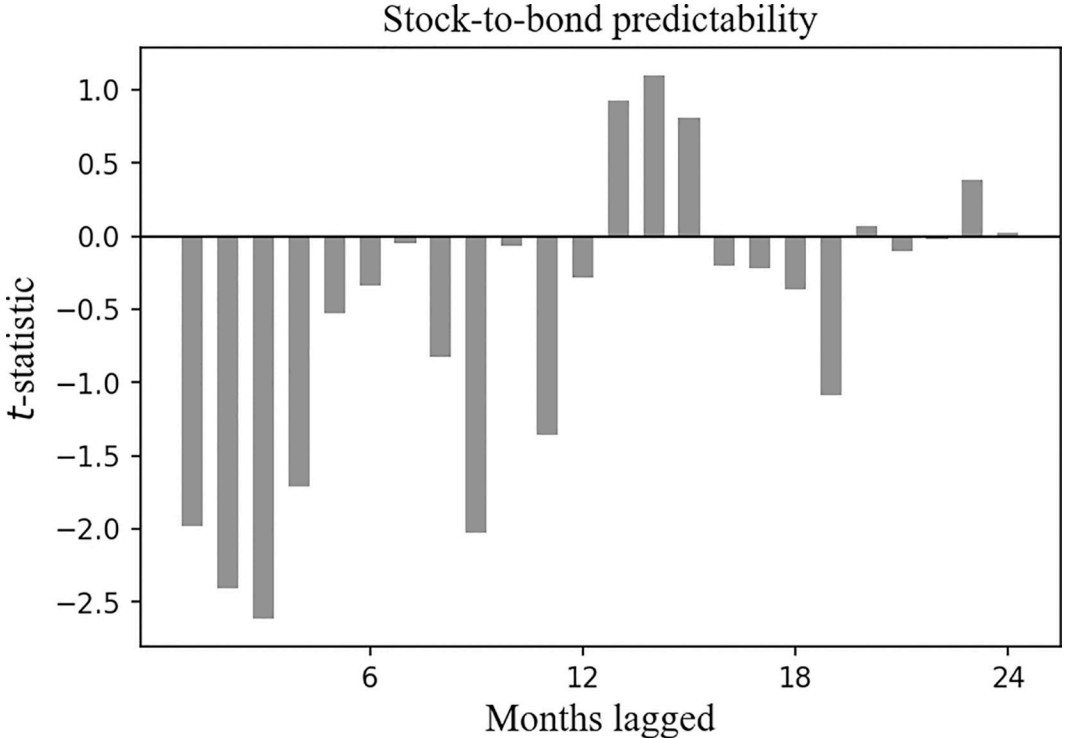

**Fig 3. Cross-asset time series predictability for the CSI 300 Index, the *t*-statistic of the $\beta_1$ clustered by month.** Plotted are the *t*-statistics of the $\beta_1$ clustered by month in Eq (1). The horizontal axis corresponds to the past month, and the vertical axis corresponds to the *t*-statistics.

In Figs 5 and 6, we report the result of Eq (2), where the horizontal axis corresponds to the past month, and the vertical axis corresponds to the *t*-statistics of the $\beta_1$ and the $\beta_2$ clustered by month.

Fig 5 reports the *t*-statistic of the $\beta_1$, and it is negative over the past 1 to 12 months. This result indicates a negative correlation between the future return of the SSE Corporate Bond Index and the historical returns of the CSI 300 Index.

In Fig 6, the *t*-statistic of the $\beta_2$ is positive over the past 1 to 3 months, which shows that the future return of the SSE Corporate Bond Index is positively correlated with its historical returns.

Also, the correlation duration between the future return of the SSE Corporate Bond Index and the historical returns of the CSI 300 Index is longer than that between the future return of the SSE Corporate Bond Index and its own historical returns.

**4.2.2. Results of cross-asset momentum with quantile regression.** In Table 3, we report the result of Eq (3). At the 5% significance level, we find that the influence of the historical cumulative returns of the CSI 300 Index on its future return is statistically significant only at the 10% quantile. The lack of significance at other quantiles suggests that the historical information of the CSI 300 Index has limited predictive power for its future return.

At the 1% significance level, we find that the influence of the historical cumulative returns of the SSE Corporate Bond Index on the future return of the CSI 300 Index is significantly positive in the 10% to 80% quantile. This result suggests that the historical information of the SSE Corporate Bond Index has a specific predictive power for the CSI 300 Index. When the SSE Corporate Bond Index has shown sustained growth, the CSI 300 Index may follow suit and increase.

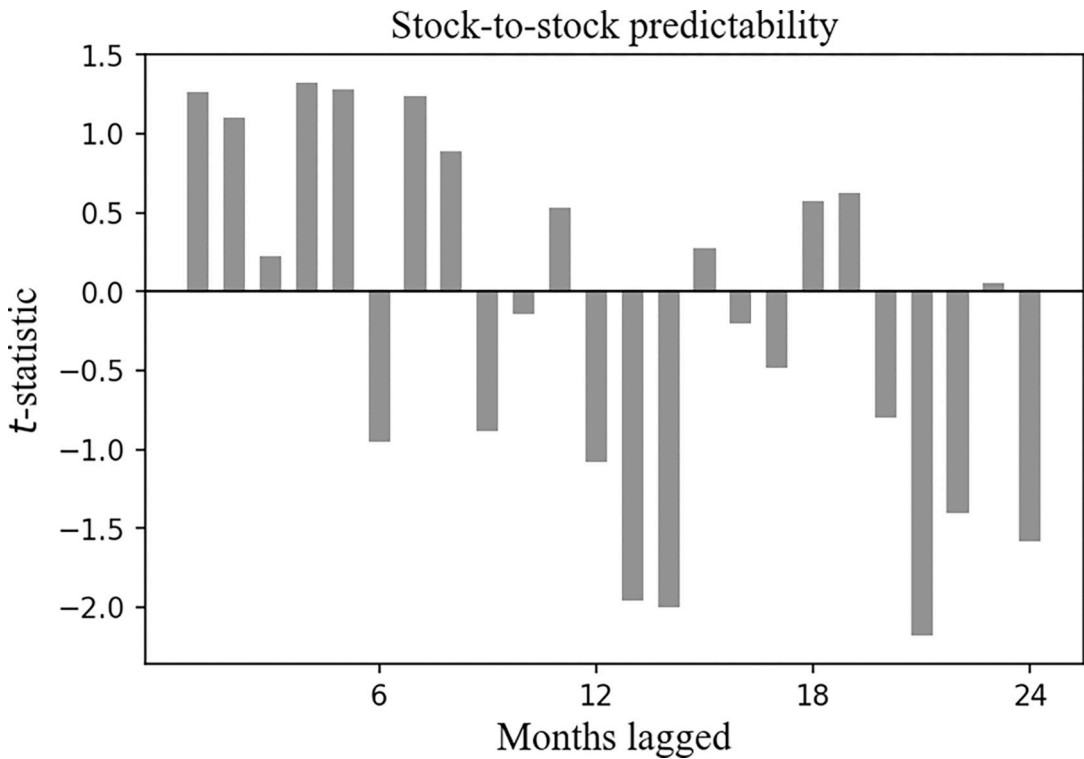

**Fig 4. Cross-asset time series predictability for the CSI 300 Index, the *t*-statistic of the $\beta_2$ clustered by month.** Plotted are the *t*-statistics of the $\beta_2$ clustered by month in Eq (1). The horizontal axis corresponds to the past month, and the vertical axis corresponds to the *t*-statistics.

In Table 4, we report the result of Eq (4). The historical cumulative returns of the SSE Corporate Bond Index significantly influence its future return only at the 80% and 90% quantiles but not at other quantiles. We can see that the historical information of the SSE Corporate Bond Index has limited predictive power for its future return.

On the other hand, at the 5% significance level, the historical cumulative returns of the CSI 300 Index significantly negatively influence the future return of the SSE Corporate Bond Index at the 10% to 60% quantile. This indicates that the historical information of the CSI 300 Index has specific predictive power for the SSE Corporate Bond Index. The SSE Corporate Bond Index may rise when the CSI 300 Index has declined continuously in the past period.

The above results suggest that the historical returns of the CSI 300 Index and the SSE Corporate Bond Index have limited predictability with short durations of influence on their future returns.

However, we observe a cross-asset momentum. The historical returns of one index have a lasting influence on the future return of the other index, and the predictability is relatively significant. Specifically, the CSI 300 Index exerts a negative cross-asset momentum effect on the SSE Corporate Bond Index, while the SSE Corporate Bond Index has a positive cross-asset momentum effect on the CSI 300 Index.

## 4.3. Results of the risk-adjusted performance of the momentum portfolio

In Table 5, we report the result of Eq (9), where columns (1) and (2) correspond to the case where short selling is allowed. In column (1), $XTSMOM_{t,t+4}$ represents the risk-return of the cross-asset momentum portfolio in Eq (8) and $Alpha_{t,t+4}$ is greater than zero at the 1%

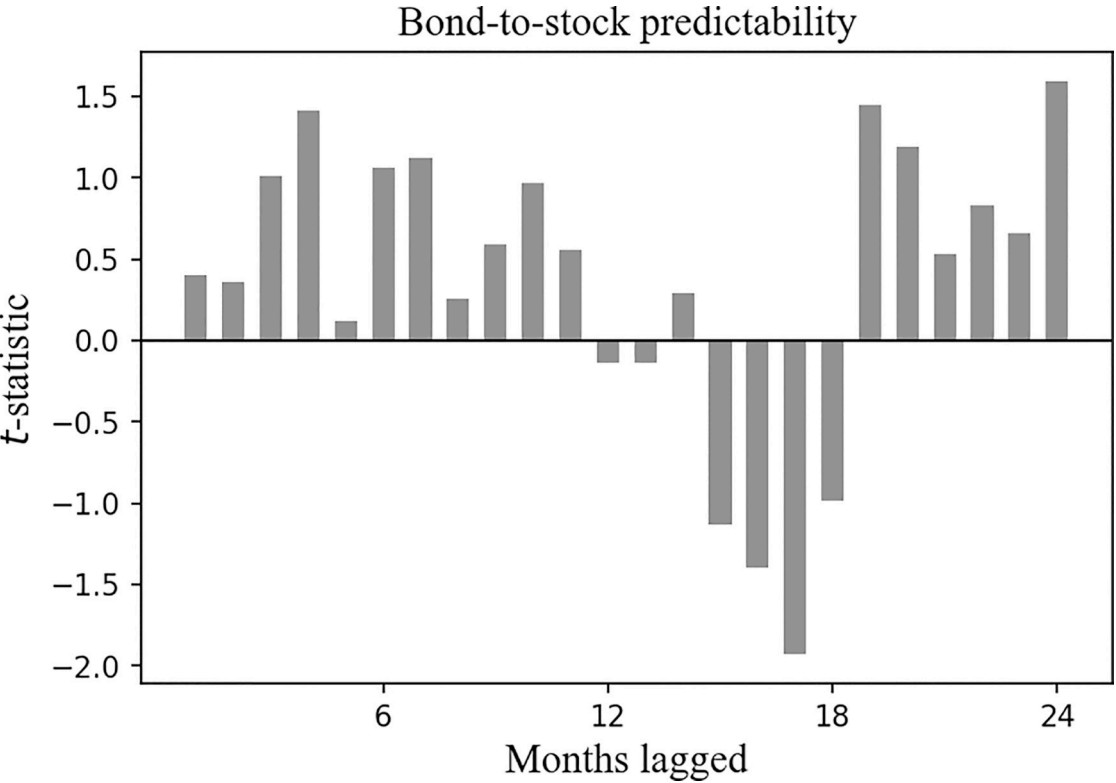

**Fig 5. Cross-asset time series predictability for the SSE Corporate Bond Index, the *t*-statistic of the $\beta_1$ clustered by month.** Plotted are the *t*-statistics of the $\beta_1$ clustered by month in Eq (2). The horizontal axis corresponds to the past month, and the vertical axis corresponds to the *t*-statistics.

significance level, with an average monthly excess return of 1.57%. In column (2), $TSMOM_{t,t+4}$ represents the risk-return of the single-asset momentum portfolio in Eq (6), $Alpha_{t,t+4}$ is greater than zero at the 5% significance level, with an average monthly excess return of 0.66%.

Columns (3) and (4) show the results for the case where short selling is prohibited. Column (3) presents $XTSMOM_{t,t+4}$ as the risk-return of the cross-asset momentum portfolio in Eq (7), with $Alpha_{t,t+4}$ greater than zero at the 5% significance level and an average monthly excess return of 0.71%. In column (4), $TSMOM_{t,t+4}$ represents the risk-return of the single-asset momentum portfolio in Eq (5), where $Alpha_{t,t+4}$ is greater than zero at the 10% significance level, with an average monthly excess return of 0.43%.

After controlling for stock pricing factors, we find that the cross-asset momentum portfolio generates significantly positive excess returns regardless of whether short selling is allowed or prohibited. Moreover, the economic and statistical significance of the excess returns is higher for the cross-asset momentum portfolio than for the single-asset momentum portfolio.

In Table 6, we report the result of Eq (10), where columns (1) and (2) correspond to the case where short selling is permitted. $XTSMOM_{t,t+4}$ represents the risk-return of the cross-asset momentum portfolio in Eq (8), and $Alpha_{t,t+4}$ is greater than zero at the 1% significance level, with an average monthly excess return of 1.18%. In column (2), $TSMOM_{t,t+4}$ represents the risk-return of the single-asset momentum portfolio in Eq (6), $Alpha_{t,t+4}$ is greater than zero at the 5% significance level, with an average monthly excess return of 0.64%.

Columns (3) and (4) show the results for the case where short selling is prohibited. Column (3) presents $XTSMOM_{t,t+4}$ as the risk-return of the cross-asset momentum portfolio in Eq (7),

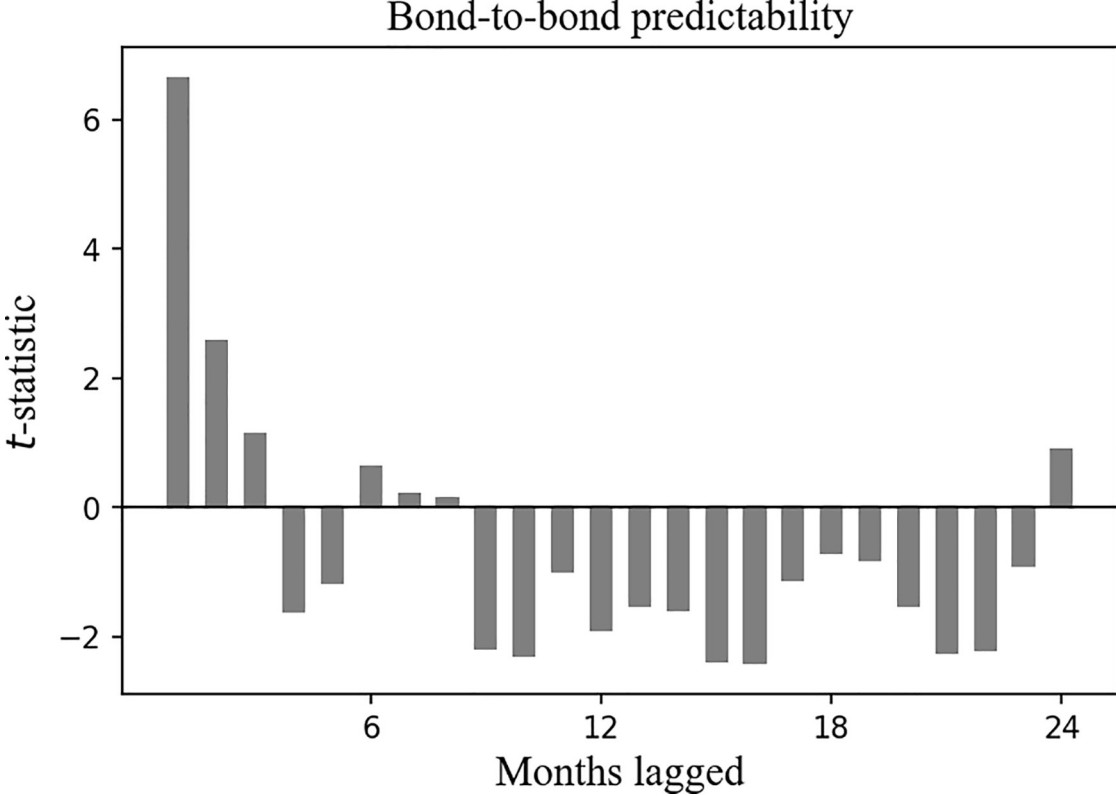

**Fig 6. Cross-asset time series predictability for the SSE Corporate Bond Index, the $t$-statistic of the $\beta_2$ clustered by month.** Plotted are the $t$-statistics of the $\beta_2$ clustered by month in Eq (2). The horizontal axis corresponds to the past month, and the vertical axis corresponds to the $t$-statistics.

with $Alpha_{t,t+4}$ greater than zero at the 10% significance level and an average monthly excess return of 0.79%. In column (4), $TSMOM_{t,t+4}$ represents the risk-return of the single-asset momentum portfolio in Eq (5), where $Alpha_{t,t+4}$ is greater than zero at the 10% significance level, with an average monthly excess return of 0.65%.

After controlling for bond pricing factors, we observe that the cross-asset momentum portfolio generates significantly positive excess returns, regardless of whether short selling is permitted or prohibited. Also, the economic and statistical significance of the excess returns of the cross-asset momentum portfolio is higher than that of the single-asset momentum portfolio.

The results in Tables 5 and 6 suggest that incorporating the cross-asset momentum into the single-asset momentum strategy can significantly enhance the portfolio's performance. After controlling for stock and bond pricing factors, the cross-asset momentum portfolio consistently outperforms the single-asset momentum portfolio, and the economic and statistical significance of the excess returns is higher for the former. Thus, these outcomes support the notion that cross-asset momentum can be a valuable addition to the existing momentum strategies.

### 4.4. Results of the cross-asset momentum transmission mechanism

In Table 7, we report the result of Eq (11) and Eq (14), in which the explained variable $FUND_{t,t+4}$ in columns (1) to (8) corresponds to the risk-return of different fund indexes in Table 1.

At the 10% significance level, the historical cumulative returns of the CSI 300 Index and the SSE Corporate Bond Index significantly influence the future return of the four hybrid fund

**Table 3. The influence of the historical accumulated returns of the market index on the future return of the CSI 300 index.**

Dependent variable: STOCK$_{t,t+4}$

| Quantile | (1) 10% | (2) 20% | (3) 30% | (4) 40% | (5) 50% |
|---|---|---|---|---|---|
| CUMSTOCK$_{t-1,t-52}$ | -0.059** | -0.020 | -0.005 | -0.009 | -0.017* |
| | (0.024) | (0.013) | (0.013) | (0.010) | (0.010) |
| CUMBOND$_{t-1,t-52}$ | 0.914*** | 0.811*** | 0.802*** | 0.833*** | 0.704*** |
| | (0.269) | (0.147) | (0.113) | (0.105) | (0.103) |
| Control variables | Yes | Yes | Yes | Yes | Yes |
| Observations | 760 | 760 | 760 | 760 | 760 |
| Quantile | (6) 60% | (7) 70% | (8) 80% | (9) 90% | |
| CUMSTOCK$_{t-1,t-52}$ | 0.004 | 0.018* | 0.016 | 0.027* | |
| | (0.010) | (0.010) | (0.011) | (0.017) | |
| CUMBOND$_{t-1,t-52}$ | 0.833*** | 0.814*** | 0.823*** | 0.858 | |
| | (0.104) | (0.100) | (0.121) | (0.179) | |
| Control variables | Yes | Yes | Yes | Yes | |
| Observations | 760 | 760 | 760 | 760 | |

This table reports the result of Eq (3). Columns (1) to (8) correspond to the dependent variable STOCK$_{t,t+4}$ at different quantiles.

Robust $t$-statistics are displayed in parentheses.

*** indicates the 1% significance level

** indicates the 5% significance level

* indicates the 10% significance level.

**Table 4. The influence of the historical accumulated returns of the market index on the future return of the SSE Corporate Bond Index.**

Dependent variable: BOND$_{t,t+4}$

| Quantile | (1) 10% | (2) 20% | (3) 30% | (4) 40% | (5) 50% |
|---|---|---|---|---|---|
| CUMSTOCK$_{t-1,t-52}$ | -0.004*** | -0.002*** | -0.001** | -0.001*** | -0.002*** |
| | (0.001) | (0.001) | (0.001) | (0.000) | (0.000) |
| CUMBOND$_{t-1,t-52}$ | -0.015 | -0.001 | -0.000 | 0.000 | -0.002 |
| | (0.016) | (0.009) | (0.006) | (0.005) | (0.005) |
| Control variables | Yes | Yes | Yes | Yes | Yes |
| Observations | 760 | 760 | 760 | 760 | 760 |
| Quantile | (6) 60% | (7) 70% | (8) 80% | (9) 90% | |
| CUMSTOCK$_{t-1,t-52}$ | -0.001*** | -0.001 | 0.000 | -0.000 | |
| | (0.001) | (0.001) | (0.001) | (0.001) | |
| CUMBOND$_{t-1,t-52}$ | -0.006 | -0.010 | -0.034*** | -0.056*** | |
| | (0.005) | (0.007) | (0.009) | (0.012) | |
| Control variables | Yes | Yes | Yes | Yes | |
| Observations | 760 | 760 | 760 | 760 | |

This table reports the result of Eq (4). Columns (1) to (8) correspond to the dependent variable BOND$_{t,t+4}$ at different quantiles.

Robust $t$-statistics are displayed in parentheses.

*** indicates the 1% significance level

** indicates the 5% significance level

* indicates the 10% significance level.

**Table 5. Risk-adjusted performance of the portfolio with stock pricing factors.**

| Dependent variable: | (1) | (2) | (3) | (4) |
|---|---|---|---|---|
| | $XTSMOM_{t,t+4}$ | $TSMOM_{t,t+4}$ | $XTSMOM_{t,t+4}$ | $tsmom_{t,t+4}$ |
| | **Long-Short** | **Long-Short** | **Long Only** | **Long Only** |
| $Alpha_{t,t+4}$ | 1.57%*** | 0.66%** | 0.71%** | 0.43%* |
| | (0.005) | (0.003) | (0.003) | (0.003) |
| $MKT_{t,t+4}$ | 0.144** | 0.088** | 0.577*** | 0.399*** |
| | (0.061) | (0.040) | (0.042) | (0.033) |
| $SMB_{t,t+4}$ | -0.387** | -0.164 | -0.230** | -0.114 |
| | (0.167) | (0.111) | (0.115) | (0.114) |
| $HML_{t,t+4}$ | 0.037 | 0.229* | 0.279** | 0.238** |
| | (0.176) | (0.117) | (0.121) | (0.096) |
| $RMW_{t,t+4}$ | -0.115 | -0.114 | -0.039 | -0.066 |
| | (0.274) | (0.184) | (0.195) | (0.150) |
| $CMA_{t,t+4}$ | 0.026 | 0.156 | 0.300* | 0.197 |
| | (0.260) | (0.171) | (0.178) | (0.141) |
| $UMD_{t,t+4}$ | -0.100 | 0.103* | 0.067 | 0.097** |
| | (0.089) | (0.060) | (0.062) | (0.049) |
| **Adjusted $R^2$** | 0.070 | 0.095 | 0.556 | 0.493 |
| **Observations** | 764 | 764 | 764 | 764 |

This table reports the result of Eq (9). Column (1) corresponds to the dependent variable XTSMOM$_{t,t+4}$ in Eq (8). Column (2) corresponds to the dependent variable TSMOM$_{t,t+4}$ in Eq (6). Column (3) corresponds to the dependent variable XTSMOM$_{t,t+4}$ in Eq (7). Column (4) corresponds to the dependent variable TSMOM$_{t,t+4}$ in Eq (5).

Robust *t*-statistics are displayed in parentheses.

*** indicates the 1% significance level

** indicates the 5% significance level

* indicates the 10% significance level.

indexes: the CSI Mixed Fund Index, the Wind Partial Stock Fund Index, the Wind Flexible Fund Index, and the Wind Partial Bond Fund Index.

Notably, the Wind Flexible Fund Index in column (6), which has the most flexible asset allocation ratio, is most significantly influenced by the historical cumulative returns of the CSI 300 Index and the SSE Corporate Bond Index. An increase of 1% in the historical cumulative returns of the stock and bond market indexes over the past 52 weeks resulted in an increase of 0.026% and 0.670%, respectively, in the following 1-month return of the Wind Flexible Fund Index.

Besides, the historical cumulative return of the CSI 300 Index does not significantly influence the CSI Stock Fund Index, the CSI Bond Fund Index, and the Wind Stock Fund Index. Also, the historical cumulative return of the SSE Corporate Bond Index has no significant influence on the Wind Bond Fund Index. Our findings suggest that the four single-asset fund indexes cannot simultaneously benefit from the momentum of the stock and bond market indexes. Hence, in the subsequent analysis of the momentum transmission mechanism, we only consider hybrid fund indexes and exclude single-asset fund indexes.

**4.4.1. Results of the momentum transmission mechanism: From the bond to the stock market.** In Table 8, we report the result of Eq (12) and Eq (13), with column (1) corresponding to Eq (12) and columns (2) to (5) corresponding to Eq (13).

In column (1), we observe that the $CUMSTOCK_{t-52,t-1}$ (the historical cumulative return of the CSI 300 Index) does not significantly influence the $STOCK_{t,t+4}$ (the future return of the

**Table 6. Risk-adjusted performance of the portfolio with bond pricing factors.**

| Dependent variable: | (1) | (2) | (3) | (4) |
|---|---|---|---|---|
| | $XTSMOM_{t,t+4}$ | $TSMOM_{t,t+4}$ | $XTSMOM_{t,t+4}$ | $TSMOM_{t,t+4}$ |
| | Long-Short | Long-Short | Long Only | Long Only |
| $Alpha_{t,t+4}$ | 1.18%** | 0.64%** | 0.79%* | 0.65%* |
| | (0.005) | (0.003) | (0.005) | (0.003) |
| $Level_{t,t+4}$ | -0.029 | -0.014 | -0.050* | -0.019 |
| | (0.027) | (0.018) | (0.026) | (0.020) |
| $Slope_{t,t+4}$ | 0.032 | 0.018 | 0.060** | 0.028 |
| | (0.029) | (0.020) | (0.029) | (0.021) |
| $Curvature_{t,t+4}$ | 0.000 | 0.000 | 0.019** | 0.016** |
| | (0.01) | (0.006) | (0.009) | (0.007) |
| Adjusted $R^2$ | 0.006 | 0.007 | 0.025 | 0.019 |
| Observations | 764 | 764 | 764 | 764 |

This table reports the result of Eq (10). Column (1) corresponds to the dependent variable $XTSMOM_{t,t+4}$ in Eq (8). Column (2) corresponds to the dependent variable $TSMOM_{t,t+4}$ in Eq (6). Column (3) corresponds to the dependent variable $XTSMOM_{t,t+4}$ in Eq (7). Column (4) corresponds to the dependent variable $TSMOM_{t,t+4}$ in Eq (5).

Robust $t$-statistics are displayed in parentheses.

*** indicates the 1% significance level

** indicates the 5% significance level

* indicates the 10% significance level.

CSI 300 Index). Consistent with the results in Table 7, we can see that the $CUMSTOCK_{t-52,t-1}$ significantly influences the future returns of the four hybrid fund indexes. These results suggest that the $CUMSTOCK_{t-52,t-1}$ may indirectly influence the $STOCK_{t,t+4}$ through the four hybrid fund indexes. We select the $CUMSTOCK_{t-52,t-1}$ as the instrumental variable for the regression of Eq (13).

Columns (2) to (5) of Table 8 show that the four hybrid fund indexes exhibit a significant intermediary effect. When the historical SSE Corporate Index rose continuously, the hybrid fund indexes amplified the rise in the CSI 300 Index. For every 1% increase in the monthly return of the CSI Mixed Fund Index, the Wind Partial Stock Fund Index, and the Wind Flexible Fund Index, the monthly return of the CSI 300 Index increases by approximately 0.733% to 0.856%. Among these three hybrid fund indexes, the Wind Flexible Fund Index, which has the most flexible asset allocation ratio, exhibits the most substantial boosting effect.

Additionally, for every 1% increase in the monthly return of the Wind Partial Bond Fund Index, the monthly return of the CSI 300 Index increases by approximately 1.875%, and this coefficient is greater than that of the other three hybrid fund indexes. This may be because the bond fund has a lower return base, resulting in a greater coefficient.

**4.4.2. Results of the momentum transmission mechanism: From the stock to the bond market.** In Table 9, we report the result of Eq (15) and Eq (16), with column (1) corresponding to Eq (15) and columns (2) to (5) corresponding to Eq (16).

In column (1), we observe that the $CUMBOND_{t-52,t-1}$ (the historical cumulative return of the SSE Corporate Bond Index) does not significantly influence the $BOND_{t,t+4}$ (the future return of the SSE Corporate Bond Index). This is consistent with the result in Table 7, which indicates that the $CUMBOND_{t-52,t-1}$ significantly influences the future returns of the four hybrid fund indexes. These results suggest that the $CUMBOND_{t-52,t-1}$ may indirectly influence the $BOND_{t,t+4}$ through the four hybrid fund indexes. We select the $CUMBOND_{t-52,t-1}$ as the instrumental variable for the regression of Eq (16).

**Table 7. The influence of the historical accumulated return of the CSI 300 index and SSE Corporate Bond Index on the future return of different fund indexes.**

Dependent variable: $FUND_{t,t+4}$

| | (1) | (2) | (3) | (4) | (5) | (6) | (7) | (8) |
|---|---|---|---|---|---|---|---|---|
| | CSI Stock Fund Index | CSI Mixed Fund Index | CSI Bond Fund Index | Wind Stock Fund Index | Wind Partial Stock Fund Index | Wind Flexible Fund Index | Wind Partial Bond Fund Index | Wind Bond Fund Index |
| $CUM$ $STOCK_{t-52,t-1}$ | 0.024 | 0.012 *** | 0.001 | 0.027 | 0.024 * | 0.026 ** | 0.011 *** | 0.001 * |
| | (0.015) | (0.005) | (0.002) | (0.017) | (0.011) | (0.014) | (0.004) | (0.002) |
| $CUM$ $BOND_{t-52,t-1}$ | 0.707 *** | 0.500 *** | 0.039 ** | 0.717 *** | 0.597 *** | 0.670 *** | 0.246 *** | 0.032 |
| | (0.131) | (0.059) | (0.019) | (0.142) | (0.128) | (0.096) | (0.040) | (0.017) |
| $SHIBOR_{t,t+4}$ | 0.001 | 0.268 | -0.181 *** | -0.015 | -0.046 | 0.078 | -0.054 | -0.200 *** |
| | (0.210) | (0.197) | (0.041) | (0.212) | (0.194) | (0.144) | (0.062) | (0.039) |
| $M2_{t,t+4}$ | 0.846 | 2.142 ** | -0.014 | 0.557 | 0.508 | 0.978 | 0.178 | -0.030 |
| | (1.109) | (0.905) | (0.191) | (1.184) | (1.066) | (0.783) | (0.305) | (0.186) |
| $CPI_{t,t+4}$ | 0.095 | -0.562 | -0.001 | -0.091 | -0.178 | 0.192 | -0.210 | -0.006 |
| | (0.636) | (0.472) | (0.089) | (0.662) | (0.615) | (0.458) | (0.201) | (0.083) |
| $PMI_{t,t+4}$ | 0.822 ** | 0.559 ** | 0.000 | 0.784 ** | 0.725 ** | 0.492 ** | 0.169 * | -0.018 |
| | (0.332) | (0.228) | (0.036) | (0.335) | (0.313) | (0.202) | (0.092) | (0.034) |
| Intercept | -2.751 *** | -0.255 | 0.197 | -2.425 ** | -2.215 ** | -2.878 *** | -0.870 *** | 0.324 ** |
| | (1.066) | (0.690) | (0.167) | (1.098) | (1.007) | (0.738) | (0.305) | (0.155) |
| Adjusted R² | 0.091 | 0.113 | 0.058 | 0.082 | 0.087 | 0.120 | 0.154 | 0.066 |
| Observations | 764 | 764 | 764 | 764 | 764 | 764 | 764 | 764 |

This table reports the results of Eq (11) and Eq (14). Columns (1) to (8) correspond to the dependent variable $FUND_{t,t+4}$ of different fund indexes.

Robust $t$-statistics are displayed in parentheses.

*** indicates the 1% significance level

** indicates the 5% significance level

* indicates the 10% significance level.

Columns (2) to (5) of Table 9 show that the four hybrid fund indexes exhibit a significant intermediary effect. When the historical CSI 300 Index rose continuously, the hybrid fund indexes exacerbated the declines in the SSE Corporate Bond Index. For every 1% increase in the monthly return of the CSI Mixed Fund Index, Wind Partial Stock Fund Index, and Wind Flexible Fund Index, the monthly return of the CSI 300 Index decreases by approximately 0.074% to 0.084%. Among these three indexes, the Wind Flexible Fund Index, which has the most flexible asset allocation ratio, is strongest in amplifying the declines.

Additionally, for every 1% increase in the monthly return of the Wind Partial Bond Fund Index, the monthly return of the SSE Corporate Bond Index decreases by approximately 0.218%.

The findings from Tables 8 and 9 reveal that the four hybrid fund indexes are intermediaries in transmitting momentum between the stock and bond markets in both directions.

The observed results are consistent with our hypothesis that when the historical price of bonds continues to rise, it increases investors' wealth level and a higher proportion of safe assets in their portfolios. As a result, investors tend to increase their investment in stocks, leading to higher future stock prices. Conversely, when the historical price of bonds declines, it

**Table 8. Result of the momentum transmission mechanism: From the bond to the stock market.**

Dependent variable: $STOCK_{t,t+4}$

| | (1) OLS | (2) IV-2SLS | (3) IV-2SLS | (4) IV-2SLS | (5) IV-2SLS |
|---|---|---|---|---|---|
| $FUND_{t,t+4}$ | | CSI Mixed Fund Index | Wind Partial Stock Fund Index | Wind Flexible Fund Index | Wind Partial Bond Fund Index |
| | | 0.733*** | 0.810*** | 0.856*** | 1.875** |
| | | (0.259) | (0.230) | (0.265) | (0.788) |
| $CUMSTOCK_{t-52,t-1}$ | -0.000 | IV | IV | IV | IV |
| | (0.009) | | | | |
| $CUMBOND_{t-52,t-1}$ | 0.685*** | 0.379*** | 0.316** | 0.347*** | 0.396** |
| | (0.082) | (0.140) | (0.131) | (0.133) | (0.156) |
| $SHIBOR_{t,t+4}$ | 0.547** | 0.271 | 0.312* | 0.207 | 0.377** |
| | (0.273) | (0.172) | (0.164) | (0.164) | (0.192) |
| $M2_{t,t+4}$ | 3.548*** | 1.403** | 1.606*** | 1.180* | 1.683** |
| | (1.257) | (0.700) | (0.606) | (0.686) | (0.765) |
| $CPI_{t,t+4}$ | 0.183 | 0.799** | 0.807** | 0.498 | 1.057*** |
| | (0.655) | (0.368) | (0.319) | (0.378) | (0.407) |
| $PMI_{t,t+4}$ | 1.014*** | 0.634** | 0.496** | 0.662** | 0.767*** |
| | (0.316) | (0.272) | (0.244) | (0.279) | (0.290) |
| Intercept | -1.597* | -2.358*** | -2.358*** | -1.688** | -2.521*** |
| | (0.957) | (0.735) | (0.656) | (0.816) | (0.742) |
| Adjusted R$^2$ | 0.114 | 0.704 | 0.773 | 0.720 | 0.617 |
| Observations | 764 | 764 | 764 | 764 | 764 |

This table reports the result of the dependent variable $STOCK_{t,t+4}$. Column (1) corresponds to Eq (12). Columns (2) to (5) correspond to Eq (13). $CUMSTOCK_{t-52,t-1}$ is the instrumental variable.

Robust $t$-statistics are displayed in parentheses.

*** indicates the 1% significance level

** indicates the 5% significance level

* indicates the 10% significance level.

leads to a decrease in investors' wealth level. As a defensive strategy, investors tend to reduce their investment in stocks, exacerbating the decline in future stock prices.

On the other hand, when the historical price of stocks continues to rise, it increases investors' wealth level. In order to seek higher returns, investors tend to reduce their investment in bonds and increase their investment in stocks, further exacerbating the decline in future bond prices. Conversely, when the historical price of stocks declines, it leads to a decrease in investors' wealth level. With risk aversion, investors tend to increase their bond investment, increasing future bond prices.

## 5. Robustness check

### 5.1. The risk-adjusted performance of the momentum portfolio after replacing the stock and bond indexes

To increase the representativeness of the stock and bond index components, we replace the CSI 300 Index and the SSE Corporate Bond Index with the SSE Composite Index and the SSE All-Bond Index. The stock index now includes all listed stocks on the Shanghai Stock Exchange, while the bond index includes treasury bonds, financial bonds, and corporate

**Table 9. Result of the momentum transmission mechanism: From the stock to the bond market.**

Dependent variable: $BOND_{t,t+4}$

| | (1)<br>OLS | (2)<br>IV-2SLS | (3)<br>IV-2SLS | (4)<br>IV-2SLS | (5)<br>IV-2SLS |
|---|---|---|---|---|---|
| $FUND_{t,t+4}$ | | CSI Mixed<br>Fund Index | Wind Partial<br>Stock<br>Fund Index | Wind Flexible<br>Fund Index | Wind Partial Bond<br>Fund Index |
| | | -0.076*** | -0.074*** | -0.084*** | -0.218*** |
| | | (0.024) | (0.024) | (0.026) | (0.073) |
| $CUMSTOCK_{t-52,t-1}$ | -0.004*** | -0.005*** | -0.006*** | -0.006*** | -0.005*** |
| | (0.000) | (0.001) | (0.001) | (0.001) | (0.001) |
| $CUMBOND_{t-52,t-1}$ | -0.004 | IV | IV | IV | IV |
| | (0.008) | | | | |
| $SHIBOR_{t,t+4}$ | -0.221*** | -0.124*** | -0.128*** | -0.117*** | -0.123*** |
| | (0.025) | (0.020) | (0.021) | (0.019) | (0.021) |
| $M2_{t,t+4}$ | -0.100 | 0.204 | 0.178 | 0.224* | 0.208 |
| | (0.116) | (0.133) | (0.136) | (0.132) | (0.132) |
| $CPI_{t,t+4}$ | 0.141** | 0.100* | 0.101* | 0.130** | 0.068 |
| | (0.060) | (0.0562) | (0.060) | (0.055) | (0.059) |
| $PMI_{t,t+4}$ | -0.006 | 0.032 | 0.039 | 0.027 | 0.021 |
| | (0.029) | (0.045) | (0.047) | (0.041) | (0.045) |
| Intercept | 0.506*** | 0.773*** | 0.794*** | 0.720*** | 0.814*** |
| | (0.088) | (0.084) | (0.090) | (0.074) | (0.094) |
| Adjusted R$^2$ | 0.146 | 0.004 | 0.109 | 0.051 | 0.057 |
| Observations | 764 | 764 | 764 | 764 | 764 |

This table reports the result of the dependent variable $BOND_{t,t+4}$. Column (1) corresponds to Eq (14). Columns (2) to (5) correspond to Eq (15). $CUMBOND_{t-52,t-1}$ is the instrumental variable.

Robust $t$-statistics are displayed in parentheses.

\*\*\* indicates the 1% significance level

\*\* indicates the 5% significance level

\* indicates the 10% significance level.

bonds from both the inter-bank market and the Shanghai & Shenzhen Stock Exchange markets. Subsequently, we reconstruct the momentum portfolio and assess its risk-adjusted performance with Eq (9) and Eq (10).

We report the results in Tables 10 and 11. Consistent with Tables 5 and 6, the cross-asset momentum portfolio exhibits a significantly positive excess return, regardless of whether short selling is allowed or prohibited. Furthermore, the excess return of the cross-asset momentum portfolio is economically and statistically significantly superior to that of the single-asset momentum portfolio.

## 5.2. The risk-adjusted performance of the momentum portfolio with different holding and lookback periods

We analyze the risk-adjusted performance of the momentum portfolio with different holding and lookback periods, as described in Eq (17):

$$r_{t,t+h} = Alpha_{t,t+h} + \beta_1 MKT_{t,t+h} + \beta_2 SMB_{t,t+h} + \beta_3 HML_{t,t+h} + \beta_4 RMW_{t,t+h} + \beta_5 CMA_{t,t+h}$$
$$+ \beta_6 UMD_{t,t+h} + \beta_7 Level_{t,t+h} + \beta_8 Slope_{t,t+h} + \beta_9 Curvature_{t,t+h} + \varepsilon_t \qquad (17)$$

**Table 10. The robustness checks: The risk-adjusted performance of the momentum portfolio with stock pricing factors.**

| Dependent variable: | (1) | (2) | (3) | (4) |
|---|---|---|---|---|
| | $XTSMOM_{t,t+4}$ | $TSMOM_{t,t+4}$ | $XTSMOM_{t,t+4}$ | $TSMOM_{t,t+4}$ |
| | **Long-Short** | **Long-Short** | **Long Only** | **Long Only** |
| $Alpha_{t,t+4}$ | 1.36%*** | 0.61%** | 0.52%* | 0.27% |
| | (0.004) | (0.003) | (0.003) | (0.002) |
| $MKT_{t,t+4}$ | 0.138** | 0.061 | 0.526*** | 0.365*** |
| | (0.055) | (0.037) | (0.037) | (0.029) |
| $SMB_{t,t+4}$ | -0.286* | -0.156 | -0.186* | -0.093 |
| | (0.152) | (0.103) | (0.103) | (0.082) |
| $HML_{t,t+4}$ | 0.015 | 0.228** | 0.295*** | 0.267*** |
| | (0.150) | (0.110) | (0.108) | (0.086) |
| $RMW_{t,t+4}$ | -0.015 | -0.145 | -0.068 | -0.119 |
| | (0.25) | (0.173) | (0.170) | (0.135) |
| $CMA_{t,t+4}$ | 0.111 | 0.063 | 0.179 | 0.050 |
| | (0.236) | (0.162) | (0.158) | (0.125) |
| $UMD_{t,t+4}$ | -0.076 | 0.086 | 0.075 | 0.091** |
| | (0.082) | (0.056) | (0.055) | (0.044) |
| **Adjusted R2** | 0.058 | 0.077 | 0.566 | 0.505 |
| **Observations** | 764 | 764 | 764 | 764 |

This table reports the empirical result of Eq (9) after replacing the stock and bond indexes. Column (1) corresponds to the dependent variable $XTSMOM_{t,t+4}$ in Eq (8). Column (2) corresponds to the dependent variable $TSMOM_{t,t+4}$ in Eq (6). Column (3) corresponds to the dependent variable $XTSMOM_{t,t+4}$ in Eq (7). Column (4) corresponds to the dependent variable $TSMOM_{t,t+4}$ in Eq (5).

Robust *t*-statistics are displayed in parentheses.

*** indicates the 1% significance level

** indicates the 5% significance level

* indicates the 10% significance level.

As presented in Table 12, we extend the holding period to 1, 3, and 6 months and shorten the lookback period to 3, 6, and 9 months.

Under the condition that short selling is allowed, the cross-asset momentum portfolio exhibits significant excess returns when the holding period is 1 month and the lookback period is 3, 6, and 9 months. When the holding period is 3 months, the cross-asset momentum portfolio still demonstrates significant excess returns with the 3-month lookback period.

On the other hand, the single-asset momentum portfolio shows significant excess returns only when the holding period is 1 month and the lookback period is 3 or 6 months. Additionally, negative excess returns are observed in the single-asset momentum portfolio with particular holding and lookback periods.

The findings are consistent under the condition that short selling is prohibited. The cross-asset momentum portfolio exhibits significant excess returns when the holding period is 1 month and the lookback period is 3, 6, and 9 months. When the holding period is 3 months, the cross-asset momentum portfolio demonstrates significant excess returns in the 3-month lookback period.

However, the single-asset momentum portfolio shows significant excess returns only when the holding period is 1 month and the lookback period is 3 months. Negative excess returns also appear in the single-asset momentum portfolio under specific holding and lookback periods.

**Table 11. The robustness checks: The risk-adjusted performance of the momentum portfolios with bond pricing factors.**

| Dependent variable: | (1) | (2) | (3) | (4) |
|---|---|---|---|---|
| | $XTSMOM_{t,t+4}$ | $TSMOM_{t,t+4}$ | $XTSMOM_{t,t+4}$ | $TSMOM_{t,t+4}$ |
| | Long-Short | Long-Short | Long Only | Long Only |
| $Alpha_{t,t+4}$ | 1.11%*** | 0.61%** | 0.71%* | 0.52%* |
| | (0.004) | (0.003) | (0.004) | (0.003) |
| $Level_{t,t+4}$ | -0.011 | -0.000 | -0.028 | -0.009 |
| | (0.024) | (-0.025) | (0.024) | (0.018) |
| $Slope_{t,t+4}$ | 0.008 | 0.003 | 0.034 | 0.015 |
| | (0.027) | (0.019) | (0.026) | (0.020) |
| $Curvature_{t,t+4}$ | -0.004 | -0.001 | 0.012 | 0.010* |
| | (0.009) | (0.006) | (0.009) | (0.006) |
| Adjusted $R^2$ | 0.003 | 0.012 | 0.019 | 0.016 |
| Observations | 764 | 764 | 764 | 764 |

This table reports the empirical result of Eq (10) after replacing the stock and bond indexes. Column (1) corresponds to the dependent variable $XTSMOM_{t,t+4}$ in Eq (8). Column (2) corresponds to the dependent variable $TSMOM_{t,t+4}$ in Eq (6). Column (3) corresponds to the dependent variable $XTSMOM_{t,t+4}$ in Eq (7). Column (4) corresponds to the dependent variable $TSMOM_{t,t+4}$ in Eq (5).

Robust $t$-statistics are displayed in parentheses.

*** indicates the 1% significance level

** indicates the 5% significance level

* indicates the 10% significance level.

These results suggest that the cross-asset momentum portfolio exhibits better robustness in terms of risk-adjusted performance compared to the single-asset momentum portfolio.

## 5.3. Cross-asset momentum transmission mechanism in different periods

We analyze the momentum transmission mechanism between the stock and bond markets in different periods, particularly in light of the rapid growth in the number and size of China's hybrid funds since 2016, as depicted in Figs 1 and 2. This growth could potentially influence the intermediary role of funds in the momentum transmission. To investigate this, we divided the sample data into two periods, around 2016.

Table 13 reports the results of Eq (12) and Eq (13) for the periods before and after 2016. Column (1) corresponds to Eq (12), while columns (2) to (5) correspond to Eq (13). Our findings reveal that before 2016, the intermediary role of the hybrid funds was not statistically significant. However, after 2016, we observed a significant intermediary effect of the hybrid funds in the momentum transmission. This suggests that the changes in the hybrid fund market in China since 2016 may influence our result, highlighting the importance of considering the evolving dynamics of the hybrid fund market.

Table 14 reports the results of Eq (15) and Eq (16) for the periods before and after 2016. Column (1) corresponds to Eq (15), while columns (2) to (5) correspond to Eq (16). The result indicates that the intermediary role of hybrid funds is statistically significant in both periods. Notably, the influence coefficient of the hybrid funds on the bond market is more pronounced after 2016, in terms of economic and statistical significance, compared to before 2016. This suggests that the influence of hybrid funds on the bond market has increased in the more recent period.

The observed results can be attributed to changes in the number and size of the hybrid funds and the underlying asymmetry in the linkage mechanism between China's stock and

**Table 12. The robustness checks: The risk-adjusted performance of the momentum portfolio with different holding and lookback periods.**

| | Long-Short | | | | | |
|---|---|---|---|---|---|---|
| | $XTSMOM_{t,t+h}$ | | | $TSMOM_{t,t+h}$ | | |
| | Holding period $h$ | | | Holding period $h$ | | |
| Lookback period $k$ | 1 month | 3 months | 6 months | 1 month | 3 months | 6 months |
| 3 months | 0.82%*** | 0.85%*** | 0.10% | 0.93%*** | 0.39% | -0.05% |
| | (0.003) | (0.003) | (0.003) | (0.003) | (0.028) | (0.029) |
| 6 months | 0.87%*** | 0.34% | 0.33% | 0.61%** | -0.01% | -0.04% |
| | (0.003) | (0.003) | (0.004) | (0.003) | (0.002) | (0.003) |
| 9 months | 0.73%** | 0.56% | 0.46% | 0.38% | -0.02% | -0.18% |
| | (0.003) | (0.004) | (0.004) | (0.003) | (0.003) | (0.003) |
| | Long Only | | | | | |
| | $XTSMOM_{t,t+h}$ | | | $TSMOM_{t,t+h}$ | | |
| | Holding period $h$ | | | Holding period $h$ | | |
| Lookback period $k$ | 1 month | 3 months | 6 months | 1 month | 3 months | 6 months |
| 3 months | 0.88%*** | 0.32%* | -0.12% | 0.60%*** | 0.01% | -0.14% |
| | (0.003) | (0.003) | (0.003) | (0.002) | (0.003) | (0.002) |
| 6 months | 0.53%** | -0.09% | -0.12% | 0.32% | -0.17% | -0.10% |
| | (0.002) | (0.003) | (0.003) | (0.002) | (0.002) | (0.002) |
| 9 months | 0.33%** | -0.11% | -0.26% | 0.08% | -0.15% | -0.35% |
| | (0.002) | (0.003) | (0.003) | (0.002) | (0.002) | (0.002) |

This table reports the results of $Alpha_{t,t+h}$ in Eq (17), with different holding and lookback periods.

Robust $t$-statistics are displayed in parentheses.

*** indicates the 1% significance level

** indicates the 5% significance level

* indicates the 10% significance level.

bond markets. Prior research by Chen and Zeng (2016) [3] has confirmed that China's stock and bond markets were relatively segregated in the past, with the linkage between the two strengthening as financial market reforms progressed. Moreover, Hou et al. (2020) [4] examined the asymmetry of risk spillovers between China's stock and bond markets and found that the intensity of risk spillovers from the stock to the bond market is more significant than vice versa. These findings align with our results presented in Tables 13 and 14. We observe that the connectivity from the stock to the bond market is more pronounced than from the bond to the stock market. Furthermore, our findings highlight the significance of the hybrid funds' intermediaries in the connectivity between the stock and bond markets.

Furthermore, considering the potential influence of the 2015 "stock crash" event on the consistency of our results, we have excluded data from 2015 and re-examined the intermediary role of funds in cross-asset momentum transmission between stocks and bonds within subsamples.

Table 15 reports the results of Eqs (12) and (13) after excluding the 2015 data. Compared to the total sample results presented in Table 8, the signs and statistical significance of the regression coefficients for the four types of hybrid funds remain consistent, with a slight increase in the coefficient magnitudes. After removing data from periods characterized by severe market volatility, the results indicate that hybrid funds continue to serve as intermediaries for bond market momentum transmission to the stock market during relatively stable market conditions.

**Table 13. The robustness checks: The momentum transmission mechanism from the bond to the stock market in different periods.**

**Before 2016**
**Dependent variable: $STOCK_{t,t+4}$**

| | (1)<br>OLS | (2)<br>IV-2SLS | (3)<br>IV-2SLS | (4)<br>IV-2SLS | (5)<br>IV-2SLS |
|---|---|---|---|---|---|
| $FUND_{t,t+4}$ | | CSI Mixed<br>Fund Index | Wind Partial<br>Stock<br>Fund Index | Wind Flexible<br>Fund Index | Wind Partial Bond<br>Fund Index |
| | | 0.062 | 0.056 | 0.229 | 0.227 |
| | | (0.625) | (0.5651) | (2.16) | (2.326) |
| $CUMSTOCK_{t-1,t-52}$ | 0.002 | IV | IV | IV | IV |
| | (0.019) | | | | |
| $CUMBOND_{t-1,t-52}$ | 0.539** | 0.507* | 0.506* | 0.470 | 0.519** |
| | (0.227) | (0.305) | (0.280) | (0.603) | (0.220) |
| Control variables | Yes | Yes | Yes | Yes | Yes |
| Adjusted R$^2$ | 0.013 | 0.103 | 0.110 | 0.217 | 0.091 |
| Observations | 334 | 334 | 334 | 334 | 334 |

**After 2016**
**Dependent variable: $STOCK_{t,t+4}$**

| | (1)<br>OLS | (2)<br>IV-2SLS | (3)<br>IV-2SLS | (4)<br>IV-2SLS | (5)<br>IV-2SLS |
|---|---|---|---|---|---|
| $FUND_{t,t+4}$ | | CSI Mixed<br>Fund Index | Wind Partial<br>Stock<br>Fund Index | Wind Flexible<br>Fund Index | Wind Partial Bond<br>Fund Index |
| | | 0.840*** | 0.944*** | 0.902*** | 2.248** |
| | | (0.280) | (0.275) | (0.276) | (0.913) |
| $CUMSTOCK_{t-1,t-52}$ | 0.022 | IV | IV | IV | IV |
| | (0.022) | | | | |
| $CUMBOND_{t-1,t-52}$ | 0.877*** | 0.330** | 0.256* | 0.342** | 0.365** |
| | (0.132) | (0.152) | (0.150) | (0.137) | (0.168) |
| Control variables | Yes | Yes | Yes | Yes | Yes |
| Adjusted R$^2$ | 0.134 | 0.747 | 0.802 | 0.763 | 0.694 |
| Observations | 436 | 436 | 436 | 436 | 436 |

This table reports the result of the dependent variable $STOCK_{t,t+4}$. Column (1) corresponds to Eq (12). Columns (2) to (5) correspond to Eq (13), and $CUMSTOCK_{t-52,t-1}$ is the instrumental variable.

Robust $t$-statistics are displayed in parentheses.

*** indicates the 1% significance level

** indicates the 5% significance level

* indicates the 10% significance level.

Table 16 reports the results of Eqs (15) and (16) after excluding the 2015 data. Similar to Table 9, which presents full sample results, the sign direction of the regression coefficients for the four types of hybrid funds remains unchanged. Some coefficients exhibit increased statistical significance and become more "negative" in value. Similar to the findings in Table 15, after excluding the 2015 data, the sub-sample results reveal that hybrid funds play a significant intermediary role in transmitting stock market momentum to the bond market.

## 5.4. Cross-asset momentum transmission mechanism with IV-GMM model

To further validate the robustness of our findings, we employ the generalized method of moments (GMM) as an alternative estimation approach in Eq (13) and Eq (16). IV-GMM

**Table 14. The robustness checks: The momentum transmission mechanism from the stock to the bond market in different periods.**

**Before 2016**
**Dependent variable: $BOND_{t,t+4}$**

| | (1) OLS | (2) IV-2SLS | (3) IV-2SLS | (4) IV-2SLS | (5) IV-2SLS |
|---|---|---|---|---|---|
| $FUND_{t,t+4}$ | | CSI Mixed Fund Index | Wind Partial Stock Fund Index | Wind Flexible Fund Index | Wind Partial Bond Fund Index |
| | | -0.049** | -0.043** | -0.085* | -0.292* |
| | | (0.022) | (0.021) | (0.044) | (0.159) |
| $CUMSTOCK_{t-52,t-1}$ | -0.002*** | -0.003*** | -0.003*** | -0.002*** | -0.004*** |
| | (0.001) | (0.001) | (0.001) | (0.001) | (0.001) |
| $CUMBOND_{t-52,t-1}$ | 0.026 | IV | IV | IV | IV |
| | (0.018) | | | | |
| Control variables | Yes | Yes | Yes | Yes | Yes |
| Adjusted $R^2$ | 0.134 | 0.747 | 0.802 | 0.763 | 0.694 |
| Observations | 436 | 436 | 436 | 436 | 436 |

**After 2016**
**Dependent variable: $BOND_{t,t+4}$**

| | (1) OLS | (2) IV-2SLS | (3) IV-2SLS | (4) IV-2SLS | (5) IV-2SLS |
|---|---|---|---|---|---|
| $FUND_{t,t+4}$ | | CSI Mixed Fund Index | Wind Partial Stock Fund Index | Wind Flexible Fund Index | Wind Partial Bond Fund Index |
| | | -0.089*** | -0.006*** | -0.097*** | -0.254*** |
| | | (0.028) | (0.002) | (0.030) | (0.087) |
| $CUMSTOCK_{t-52,t-1}$ | -0.009*** | -0.006*** | -0.088*** | -0.006*** | -0.006*** |
| | (0.001) | (0.002) | (0.029) | (0.002) | (0.002) |
| $CUMBOND_{t-52,t-1}$ | -0.058 | IV | IV | IV | IV |
| | (0.043) | | | | |
| Control variables | Yes | Yes | Yes | Yes | Yes |
| Adjusted $R^2$ | 0.134 | 0.747 | 0.802 | 0.763 | 0.694 |
| Observations | 436 | 436 | 436 | 436 | 436 |

This table reports the result of the dependent variable $BOMD_{t,t+4}$. Column (1) corresponds to Eq (14). Columns (2) to (5) correspond to Eq (15), and $CUMBOND_{t-52,t-1}$ is the instrumental variable.

Robust $t$-statistics are displayed in parentheses.

*** indicates the 1% significance level

** indicates the 5% significance level

* indicates the 10% significance level.

model helps address potential issues of heteroscedasticity or autocorrelation in the time series data, and can provide more efficient estimates compared to 2SLS.

Tables 17 and 18 report the results obtained from the IV-GMM model. Consistent with the results in Table 8 and Table 9, we reaffirm that hybrid funds play an intermediary role in the momentum transmission mechanism between the stock and bond markets. These results further support the robustness and reliability of our findings, reinforcing the significance of hybrid funds in facilitating the transmission of cross-asset momentum.

## 6. Conclusion

We investigate the cross-asset momentum between China's stock and bond markets and the role of the hybrid funds in the transmission mechanism of cross-asset momentum. Our

**Table 15. The robustness checks: The momentum transmission mechanism from the bond to the stock market, excluding data from 2015.**

Exclude 2015
Dependent variable: $STOCK_{t,t+4}$

| | (1) OLS | (2) IV-2SLS | (3) IV-2SLS | (4) IV-2SLS | (5) IV-2SLS |
|---|---|---|---|---|---|
| $FUND_{t,t+4}$ | | CSI Mixed Fund Index | Wind Partial Stock Fund Index | Wind Flexible Fund Index | Wind Partial Bond Fund Index |
| | | 0.799*** | 0.864*** | 0.923*** | 2.017*** |
| | | (0.273) | (0.268) | (0.297) | (0.801) |
| $CUMSTOCK_{t-52,t-1}$ | 0.001 | IV | IV | IV | IV |
| | (0.019) | | | | |
| $CUMBOND_{t-52,t-1}$ | 0.710*** | 0.422*** | 0.374*** | 0.420*** | 0.433** |
| | (0.091) | (0.174) | (0.149) | (0.122) | (0.151) |
| Control variables | Yes | Yes | Yes | Yes | Yes |
| Adjusted $R^2$ | 0.022 | 0.191 | 0.214 | 0.331 | 0.097 |
| Observations | 712 | 712 | 712 | 712 | 712 |

This table reports the result of the dependent variable $STOCK_{t,t+4}$. Column (1) corresponds to Eq (12). Columns (2) to (5) correspond to Eq (13), and $CUMSTOCK_{t-52,t-1}$ is the instrumental variable.

Robust *t*-statistics are displayed in parentheses.

*** indicates the 1% significance level

** indicates the 5% significance level

* indicates the 10% significance level.

**Table 16. The robustness checks: The momentum transmission mechanism from the stock to the bond market, excluding data from 2015.**

Exclude 2015
Dependent variable: $BOND_{t,t+4}$

| | (1) OLS | (2) IV-2SLS | (3) IV-2SLS | (4) IV-2SLS | (5) IV-2SLS |
|---|---|---|---|---|---|
| $FUND_{t,t+4}$ | | CSI Mixed Fund Index | Wind Partial Stock Fund Index | Wind Flexible Fund Index | Wind Partial Bond Fund Index |
| | | -0.083*** | -0.081*** | -0.091*** | -0.313* |
| | | (0.017) | (0.019) | (0.038) | (0.127) |
| $CUMSTOCK_{t-52,t-1}$ | -0.005*** | -0.006*** | -0.007*** | -0.007*** | -0.006*** |
| | (0.001) | (0.001) | (0.001) | (0.001) | (0.001) |
| $CUMBOND_{t-52,t-1}$ | 0.029 | IV | IV | IV | IV |
| | (0.021) | | | | |
| Control variables | Yes | Yes | Yes | Yes | Yes |
| Adjusted $R^2$ | 0.198 | 0.821 | 0.875 | 0.824 | 0.749 |
| Observations | 712 | 712 | 712 | 712 | 712 |

This table reports the result of the dependent variable $BOND_{t,t+4}$. Column (1) corresponds to Eq (14). Columns (2) to (5) correspond to Eq (15), and $CUMBOND_{t-52,t-1}$ is the instrumental variable.

Robust *t*-statistics are displayed in parentheses.

*** indicates the 1% significance level

** indicates the 5% significance level

* indicates the 10% significance level.

**Table 17. The robustness checks: The result of the momentum transmission mechanism with IV-GMM, from the bond to the stock market.**

Dependent variable: $STOCK_{t,t+4}$
Instrumental variable: $CUMSTOCK_{t-52,t-1}$

| $FUND_{t,t+4}$ | (1) CSI Mixed Fund Index | (2) Wind Partial Stock Fund Index | (3) Wind Flexible Fund Index | (4) Wind Partial Bond Fund Index |
|---|---|---|---|---|
|  | 0.773*** | 0.860*** | 0.912*** | 2.163** |
|  | (0.263) | (0.240) | (0.267) | (0.860) |
| $CUMBOND_{t-52,t-1}$ | 0.373*** | 0.305** | 0.338*** | 0.382** |
|  | (0.136) | (0.129) | (0.127) | (0.151) |
| $SHIBOR_{t,t+4}$ | 0.198 | 0.234 | 0.117 | 0.189 |
|  | (0.163) | (0.154) | (0.156) | (0.155) |
| $M2_{t,t+4}$ | 1.214* | 1.408** | 0.943 | 1.197 |
|  | (0.691) | (0.596) | (0.683) | (0.754) |
| $CPI_{t,t+4}$ | 0.806** | 0.816*** | 0.487 | 1.118*** |
|  | (0.356) | (0.308) | (0.364) | (0.384) |
| $PMI_{t,t+4}$ | 0.616** | 0.468* | 0.643** | 0.737*** |
|  | (0.266) | (0.242) | (0.271) | (0.277) |
| Intercept | -2.499*** | -2.514*** | -1.810** | -2.939*** |
|  | (0.689) | (0.619) | (0.754) | (0.632) |
| Adjusted R$^2$ | 0.722 | 0.784 | 0.742 | 0.661 |
| Observations | 764 | 764 | 764 | 764 |

This table reports the result of the dependent variable $STOCK_{t,t+4}$. Columns (1) to (4) correspond to Eq (13) with the IV-GMM model. $CUMSTOCK_{t-52,t-1}$ is the instrumental variable.

Robust $t$-statistics are displayed in parentheses.

*** indicates the 1% significance level

** indicates the 5% significance level

* indicates the 10% significance level.

findings reveal a significant cross-asset momentum effect, characterized by a negative correlation between the stock market's historical momentum and the bond market's future return, and a positive correlation between the bond market's historical momentum and the stock market's future return.

Furthermore, we construct the stock-bond index portfolio based on cross-asset momentum, demonstrating a significantly positive excess return that other investment factors cannot explain. Moreover, the excess return of the cross-asset momentum portfolio outperforms that of the single-asset momentum portfolio, indicating the robustness of our findings.

Our analysis also highlights the intermediary role of the hybrid funds in the cross-asset momentum transmission mechanism. Specifically, when the stock market exhibits upward momentum, hybrid funds exacerbate the decline in bond prices. Conversely, when upward momentum exists in the bond market, hybrid funds exacerbate the rise in stock prices. Furthermore, we find that the intermediary effect of the hybrid funds is strengthened by the flexibility of their asset allocation ratio, and this effect has become more pronounced with the growth of the number and size of the hybrid funds in China in recent years.

The findings of our study contribute to affirming the positive role of the hybrid fund in enhancing the liquidity of China's financial market and improving pricing efficiency. Our results also carry important policy implications for emerging markets. First, relaxing restrictions on the fund's asset allocation ratio and expanding its investment scope could stimulate

**Table 18. The robustness checks: The result of the momentum transmission mechanism with IV-GMM, from the stock to the bond market.**

Dependent variable: $BOND_{t,t+4}$
Instrumental variable: $CUMBOND_{t-52,t-1}$

| $FUND_{t,t+4}$ | (1)<br>CSI Mixed<br>Fund Index | (2)<br>Wind Partial<br>Stock<br>Fund Index | (3)<br>Wind Flexible<br>Fund Index | (4)<br>Wind Partial Bond<br>Fund Index |
|---|---|---|---|---|
|  | -0.076*** | -0.074*** | -0.084*** | -0.218*** |
|  | (0.024) | (0.024) | (0.026) | (0.073) |
| $CUMSTOCK_{t-52,t-1}$ | -0.005*** | -0.006*** | -0.006*** | -0.005*** |
|  | (0.001) | (0.001) | (0.001) | (0.001) |
| $SHIBOR_{t,t+4}$ | -0.124*** | -0.128*** | -0.117*** | -0.123*** |
|  | (0.020) | (0.021) | (0.019) | (0.021) |
| $M2_{t,t+4}$ | 0.204 | 0.178 | 0.224* | 0.208 |
|  | (0.133) | (0.136) | (0.132) | (0.132) |
| $CPI_{t,t+4}$ | 0.100* | 0.101* | 0.130** | 0.068 |
|  | (0.056) | (0.060) | (0.055) | (0.059) |
| $PMI_{t,t+4}$ | 0.032 | 0.039 | 0.027 | 0.021 |
|  | (0.045) | (0.047) | (0.041) | (0.045) |
| Intercept | 0.773*** | 0.794*** | 0.720*** | 0.814*** |
|  | (0.084) | (0.090) | (0.074) | (0.094) |
| Adjusted R$^2$ | 0.031 | 0.109 | 0.051 | 0.057 |
| Observations | 764 | 764 | 764 | 764 |

This table reports the result of the dependent variable $BOND_{t,t+4}$. Columns (1) to (4) correspond to Eq (15) with the IV-GMM model. $CUMBOND_{t-52,t-1}$ is the instrumental variable.

Robust $t$-statistics are displayed in parentheses.

*** indicates the 1% significance level

** indicates the 5% significance level

* indicates the 10% significance level.

the vitality of the fund industry and promote its high-quality development. Second, encouraging the development of the hybrid fund could improve the efficiency of capital and information flows between the stock and bond markets, reduce the impact of single-market risk, and better meet investors' evolving wealth management needs.

## Supporting information

**S1 Appendix.**
(DOCX)

**S1 Dataset.**
(XLSX)

## Author Contributions

**Conceptualization:** Xiaowei Wang.

**Data curation:** Xiaowei Wang.

**Formal analysis:** Xiaowei Wang.

**Investigation:** Xiaowei Wang.

**Methodology:** Xiaowei Wang.

**Project administration:** Xiaowei Wang.

**Resources:** Xiaowei Wang.

**Software:** Xiaowei Wang.

**Supervision:** Rui Wang, Yichun Zhang.

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
