## [Decision Letter · Decision Letter 0]

25 Sep 2023

PONE-D-23-23069Cross-asset momentum and the hybrid fund transmission mechanism in China's stock and bond marketsPLOS ONE

Dear Dr. Wang,

Thank you for submitting your manuscript to PLOS ONE. After careful consideration, we feel that it has merit but does not fully meet PLOS ONE’s publication criteria as it currently stands. Therefore, we invite you to submit a revised version of the manuscript that addresses the points raised during the review process.

We look forward to receiving your revised manuscript.

Kind regards,

Hung Do

Academic Editor

PLOS ONE

3. Please remove your figures from within your manuscript file, leaving only the individual TIFF/EPS image files, uploaded separately. These will be automatically included in the reviewers’ PDF.

Additional Editor Comments:

I have now received evaluations from two referees. They both find the potential of the paper but also raise some major concerns, including the motivation and the mechanism of the relationship. Please find detail in the referees' reports and carefully address them point by point. We look forward to receiving the revised version of the paper.

Reviewers' comments:

Reviewer's Responses to Questions

**Comments to the Author**

1. Is the manuscript technically sound, and do the data support the conclusions?

Reviewer #1: Partly

Reviewer #2: Yes

2. Has the statistical analysis been performed appropriately and rigorously? 

Reviewer #1: Yes

Reviewer #2: Yes

3. Have the authors made all data underlying the findings in their manuscript fully available?

Reviewer #1: Yes

Reviewer #2: No

4. Is the manuscript presented in an intelligible fashion and written in standard English?

Reviewer #1: Yes

Reviewer #2: Yes

5. Review Comments to the Author

Reviewer #1: The authors explore a cross-asset momentum effect between the stock and bond markets in China. This question is interesting, and the empirical results partially support the conclusions. The comments as follows:

(1) The introduction needs to explain why this research is necessary? Is it merely because China’s capital market has become one of the largest in the world? Are there other theoretical implications?

(2) In 2015, there was a “stock crash” event in China stock market. The robustness check should analyze whether the conclusions are consistent by excluding this period?

(3) I think the subscript of sigma in equation (1) should be (t-51,t-1), not (t-1,t-51).

(4) why are the variables the ratio of asset-return to volatility in equations (1)-(2)? Why not asset-return? Also, should the explanatory variable contain all lag terms, i.e. \\sum_{k=1}^{n}{\\beta_k\\frac{BOND_{t-k,t+4-k}}{\\sigma_{t-52-k,t-1-k}^{BOND}}}?

(5) Equation (8) shows the cross-asset momentum portfolio. However, as I known, it should compare the return between different assets when constructing a portfolio based on momentum effect, rather than comparing the returns to zero.

(6) In equation (9), _,+4 represents the market risk factor. Where is the “market” in here, the stock market, or the bond market?

(7) the authors point out that the equations (11) to (13) can investigate the intermediary role of hybrid funds. I don’t think that is appropriate. Because it is possible that the trading by other investors influence the prices of stocks and bonds, and then the return of hybrid fund changes but it may be not trade at all.

Overall, the mechanism analysis in this paper is flawed. The empirical results demonstrate the cross-asset momentum, but the authors do not delve into the reason of this phenomenon. The innovation of this paper is meager.

Reviewer #2: Review Report

Paper: Cross-asset momentum and the hybrid fund transmission mechanism in

China’s stock and bond markets

Comments:

The idea of the paper is interesting, the paper is quite well-written. However, there are several comments:

- The authors used many arguments without any reference, for example: “July 2022,

the total market capitalization of China’s A-share market reached approximately 12

trillion US dollars, making it one of the largest stock markets in the world. Additionally,

the bond market size in China has grown substantially, reaching up to 20 trillion US

dollars, becoming the second largest bond market globally”, and many other arguments in the Introduction parts.

- The data sources for all figures need to be cited.

- I cannot see any content of your Panels A, B in the introduction part. Are they the Figures 1A, 1B at the end of your paper?

- The authors should structure the Introduction part: There is no findings, implication here.

6. PLOS authors have the option to publish the peer review history of their article (what does this mean?). If published, this will include your full peer review and any attached files.

Reviewer #1: No

Reviewer #2: No

---

## [Author Response · Author response to Decision Letter 0]

8 Nov 2023

Dear Editors and Reviewers,

Due to space constraints, we have placed our responses to the suggested revisions in an attachment titled 'Response to Reviewers.' For further details, please refer to the attached document.

Best regards

---

## [Decision Letter · Decision Letter 1]

10 Jan 2024

PONE-D-23-23069R1Cross-asset momentum and the hybrid fund transmission mechanism in China's stock and bond marketsPLOS ONE

Dear Dr. Wang,

Thank you for submitting your manuscript to PLOS ONE. After careful consideration, we feel that it has merit but does not fully meet PLOS ONE’s publication criteria as it currently stands. Therefore, we invite you to submit a revised version of the manuscript that addresses the points raised during the review process.

We look forward to receiving your revised manuscript.

Kind regards,

Hung Do

Academic Editor

PLOS ONE

Additional Editor Comments:

The authors have addressed most of concerns raised, however, there is one important point left regarding the intermediary role of the hybrid funds. Please find detail in the comment of the reviewer 1.

Reviewers' comments:

Reviewer's Responses to Questions

**Comments to the Author**

1. If the authors have adequately addressed your comments raised in a previous round of review and you feel that this manuscript is now acceptable for publication, you may indicate that here to bypass the “Comments to the Author” section, enter your conflict of interest statement in the “Confidential to Editor” section, and submit your "Accept" recommendation.

Reviewer #1: (No Response)

Reviewer #2: All comments have been addressed

2. Is the manuscript technically sound, and do the data support the conclusions?

Reviewer #1: Partly

Reviewer #2: Yes

3. Has the statistical analysis been performed appropriately and rigorously? 

Reviewer #1: Yes

Reviewer #2: Yes

4. Have the authors made all data underlying the findings in their manuscript fully available?

Reviewer #1: Yes

Reviewer #2: Yes

5. Is the manuscript presented in an intelligible fashion and written in standard English?

Reviewer #1: Yes

Reviewer #2: Yes

6. Review Comments to the Author

Reviewer #1: The authors revised the paper based on comments, but I don't think that equations (11) to (13) can fully explain the intermediary role of hybrid funds. For instance, in one hybrid fund, 60% of its total assets are invested in stocks. If these stocks yield a return of 50%, then the hybrid fund will generate a return of 30%. Since stock prices impact returns for hybrid funds due to their inclusion as components within them, it seems rather intuitive. Therefore, testing the intermediary effect of hybrid fund returns is meaningless; instead, examining shareholding ratios may be more persuasive.

Reviewer #2: (No Response)

7. PLOS authors have the option to publish the peer review history of their article (what does this mean?). If published, this will include your full peer review and any attached files.

Reviewer #1: No

Reviewer #2: No

---

## [Author Response · Author response to Decision Letter 1]

24 Jan 2024

Dear Editor and Reviewers,

As per your request, we have uploaded the "Manuscript," "Revised Manuscript with Track Changes," and "Response to Reviewers" files onto the system.

In our detailed "Response to Reviewers," we have addressed the concerns raised by the reviewers. Additionally, modifications made to the manuscript are highlighted in red in the "Revised Manuscript with Track Changes."

Kindly review the attached documents for your perusal. 

Thank you!

Best regards

---

## [Decision Letter · Decision Letter 2]

6 Mar 2024

Cross-asset momentum and the hybrid fund transmission mechanism in China's stock and bond markets

PONE-D-23-23069R2

Dear Dr. Wang,

We’re pleased to inform you that your manuscript has been judged scientifically suitable for publication and will be formally accepted for publication once it meets all outstanding technical requirements.

Kind regards,

Hung Do

Academic Editor

PLOS ONE

Additional Editor Comments (optional):

Reviewers' comments:

Reviewer's Responses to Questions

**Comments to the Author**

1. If the authors have adequately addressed your comments raised in a previous round of review and you feel that this manuscript is now acceptable for publication, you may indicate that here to bypass the “Comments to the Author” section, enter your conflict of interest statement in the “Confidential to Editor” section, and submit your "Accept" recommendation.

Reviewer #1: All comments have been addressed

2. Is the manuscript technically sound, and do the data support the conclusions?

Reviewer #1: Yes

3. Has the statistical analysis been performed appropriately and rigorously? 

Reviewer #1: Yes

4. Have the authors made all data underlying the findings in their manuscript fully available?

Reviewer #1: Yes

5. Is the manuscript presented in an intelligible fashion and written in standard English?

Reviewer #1: Yes

6. Review Comments to the Author

Reviewer #1: The authors responded my comments and revised the paper detail. I am satisfied with the response. However, there is stall a shortcoming. Theoretically, the relationship between rate and stock price is negative, which means that bond and stock returns are positively correlated. Therefore, the conclusion that the momentum in the bond market positively influences the stock market’s return is obvious. Is this momentum effect? I would like the authors to explain this question.

7. PLOS authors have the option to publish the peer review history of their article (what does this mean?). If published, this will include your full peer review and any attached files.

Reviewer #1: No

---

## [Editor Report · Acceptance letter]

11 Mar 2024

PONE-D-23-23069R2 

PLOS ONE

Dear Dr. Wang, 

I'm pleased to inform you that your manuscript has been deemed suitable for publication in PLOS ONE. Congratulations! Your manuscript is now being handed over to our production team.

Kind regards, 

on behalf of

Dr. Hung Do 

Academic Editor

PLOS ONE